# CURVATURE-GUIDED DYNAMIC SCALE NETWORKS FOR MULTI-VIEW STEREO

**Khang Truong Giang**[a]    **Soohwan Song**[b]    **Sungho Jo**[a]
[a] School of Computing, KAIST, Daejeon, 34141, Republic of Korea
[b] Intelligent Robotics Research Division, ETRI, Daejeon 34129, Republic of Korea
{khangtg,shjo}@kaist.ac.kr, soohwansong@etri.re.kr

## ABSTRACT

Multi-view stereo (MVS) is a crucial task for precise 3D reconstruction. Most recent studies tried to improve the performance of matching cost volume in MVS by designing aggregated 3D cost volumes and their regularization. This paper focuses on learning a robust feature extraction network to enhance the performance of matching costs without heavy computation in the other steps. In particular, we present a dynamic scale feature extraction network, namely, CDSFNet. It is composed of multiple novel convolution layers, each of which can select a proper patch scale for each pixel guided by the normal curvature of the image surface. As a result, CDFSNet can estimate the optimal patch scales to learn discriminative features for accurate matching computation between reference and source images. By combining the robust extracted features with an appropriate cost formulation strategy, our resulting MVS architecture can estimate depth maps more precisely. Extensive experiments showed that the proposed method outperforms other methods on complex outdoor scenes. It significantly improves the completeness of reconstructed models. As a result, the method can process higher resolution inputs within faster run-time and lower memory than other MVS methods.

## 1 INTRODUCTION

A challenging problem in multi-view stereo (MVS) is accurately estimating dense correspondences across a collection of high-resolution images. Many MVS studies have tried to solve ill-posed MVS problems such as matching ambiguity and high computational complexity. In such problems, learning-based MVS methods (Chen et al., 2019; Luo et al., 2019; Chen et al., 2020a; Xu & Tao, 2020a; Xue et al., 2019) usually outperform traditional methods (Galliani et al., 2015; Schonberger & Frahm, 2016). The learning-based methods are generally composed of three steps, including feature extraction, cost volume formulation, and cost volume regularization. Most studies made an effort to improve the performance of cost formulation (Luo et al., 2019; Zhang et al., 2020; Xu & Tao, 2020c; Yi et al., 2020) or cost regularization (Luo et al., 2019; 2020; Wang et al., 2021). The visibility information (Zhang et al., 2020; Xu & Tao, 2020c) or attention techniques (Yi et al., 2020; Luo et al., 2020) are considered for cost aggregation and feature matching, respectively. Furthermore, some studies employed a hybrid 3D UNet structure (Luo et al., 2019; Sormann et al., 2020) for cost volume regularization. While these approaches significantly improve MVS performances, their feature extraction networks have some drawbacks that degrade the quality of 3D reconstruction. First, they observed a restricted size of receptive fields from the fixed size of convolutional kernels. This leads to difficulty in learning a robust pixel-level representation when an object's scale varies extremely in images. As a result, the extracted features cause the low quality of matching costs, especially when the difference of camera poses between reference and source images is large. Second, MVS networks are often trained on low-resolution images because of limited memory and restricted computation time. Therefore, the existing feature extraction methods utilizing fixed-scale feature representation on deep networks could not generalize to high-resolution images in prediction; the MVS performance is downgraded on a high-resolution image set.

To address these issues, we propose a new feature extraction network that can be adapted to various object scales and image resolutions. We refer the proposed network as curvature-guided dynamic

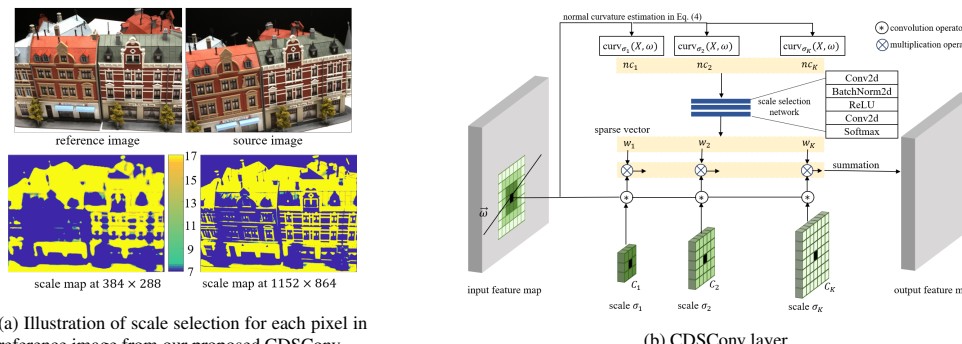

(a) Illustration of scale selection for each pixel in reference image from our proposed CDSConv

(b) CDSConv layer

Figure 1: The results extracted by the proposed CDSConv and overall pipeline of CDSConv

scale feature network (CDSFNet). The proposed network is composed of novel convolution layers, curvature-guided dynamic scale convolution (CDSConv), which select a suitable patch scale for each pixel to learn robust representation. This selection is achieved by computing the normal curvature of the image surface at several candidate scales and then analyzing the computed outputs via a classification network. The pixel-level patch scales estimated from the proposed network are dynamic with respect to textures, scale of objects, and epipolar geometry. Therefore, it trains more discriminative features than existing networks (Gu et al., 2020; Cheng et al., 2020; Yang et al., 2020; Zhang et al., 2020; Luo et al., 2020) for accurate matching cost computation. Fig. 1a illustrates the dynamic pixel-level patch scales estimated from CDSConv where the window size is ranged from $7 \times 7$ to $17 \times 17$. As shown in the figure, CDSConv predicts the adaptive patch scales where the scale is small in thin structure or texture-rich regions and large in the texture-less regions. Moreover, the scale is dynamic according to image resolution. CDSConv generally produces small scale values in a low-resolution image and large scale values in a high-resolution image.

We also formulate a new MVS framework, CDS-MVSNet, which aims at improving the quality of MVS reconstruction while decreasing computation time and memory consumption. Similar to previous works (Gu et al., 2020; Cheng et al., 2020; Yang et al., 2020; Wang et al., 2021), CDS-MVSNet is composed of a cascade network structure to estimate high-resolution depth maps in a coarse-to-fine manner. For each cascade stage, it formulates a 3D cost volume based on the output features of CDSFNet, which reduces the matching ambiguity by using proper pixel's scale. For instance, consider a 3D point on the scene that needs to be reconstructed, CDSFNet can capture the same context information in both reference and source views for this point to extract the matchable features. Furthermore, CDS-MVSNet applies visibility-based cost aggregation to improve the performance of stereo matching. The pixel-wise visibility is estimated from the curvature information that encodes the matching ability of extracted features implicitly. Therefore, CDS-MVSNet performs accurate stereo matching and cost aggregation even on low-resolution images. Finally, the proposed network can generate high quality 3D models only by processing half-resolution images. This approach can significantly reduce the computation time and memory consumption of MVS reconstruction.

The contribution of this paper is summarized as follows:

- We present the CDSConv that learns dynamic scale features guided by the normal curvature of image surface. This operation is implemented by approximating surface normal curvatures in several candidate scales and choosing a proper scale via a classification network.

- We propose a new feature extraction network, CDSFNet, which is composed of multiple CDSConv layers for learning robust representation at pixel level. CDFSNet estimates the optimal pixel's scale to learn features. The scale is selected adaptively with respect to structures, textures, and epipolar constraints.

- We present CDS-MVSNet for MVS depth estimation. CDS-MVSNet performs accurate stereo matching and cost aggregation by handling the ambiguity and visibility in the image matching process. It also significantly reduces the run-time and memory consumption by processing the half-resolution images while maintaining the reconstruction quality.

- We verify the effectiveness of our method on two benchmark datasets; DTU (Aanæs et al., 2016) and Tanks & Temples (Knapitsch et al., 2017). The results demonstrate that the proposed feature learning method boosts the performance of MVS reconstruction.

## 2 RELATED WORK

**Multi-view Stereo.** Traditional MVS studies can be divided into three categories: volumetric, point cloud-based, and depth map-based methods (Furukawa & Hernández, 2015). Comparatively, depth map-based methods are more concise and flexible, they estimate depth maps for all reference images and then perform depth fusion to achieving a 3D model (Galliani et al., 2015; Schönberger et al., 2016; Xu & Tao, 2019). The popular MVS methods in this category are usually based on Patch-Match Stereo algorithm (Barnes et al., 2009). Recently, many studies have applied a learning-based approach, which has achieved significant performance improvements over traditional methods. Several volumetric methods (Kar et al., 2017; Ji et al., 2017) employed 3D CNNs to explicitly predict global 3D surfaces. Yao et al. (2018) proposed a depth-map-based method, MVSNet, which is an end-to-end trainable architecture with three steps. However, these methods still confronted the matching ambiguity problem and could not process high-resolution images due to limited memory. Many methods based on the three-step architecture of MVSNet are proposed to address these issues. Several studies (Gu et al., 2020; Cheng et al., 2020; Yang et al., 2020; Wang et al., 2021; Zhang et al., 2020) applied a coarse-to-fine framework which estimates the depths through multiple stages to reduce the computational complexity. They used the estimated depth of each stage to adjust the depth hypothesis range for cost formulation in the next stage. To enhance the matching cost volume, other methods introduced a novel strategy for cost volume formulation. Luo et al. (2019) proposed a patch-wise feature matching instead of pixel-wise matching to compute the cost. Yi et al. (2020) and Luo et al. (2020) applied an attention technique to improve feature matching. Zhang et al. (2020); Xu & Tao (2020c) estimated visibility information to score the view weights for cost aggregation.

All these methods (Gu et al., 2020; Cheng et al., 2020; Yang et al., 2020; Wang et al., 2021; Zhang et al., 2020; Luo et al., 2020; Yu & Gao, 2020) had a trade-off between computation and performance; the methods with high performance usually require expensive computations. Also, they only applied standard CNNs with static scale representation for feature extraction. In this work, we aim to learn dynamic scale representation for more accurate feature matching. We also estimate pixel-wise visibility information to remove noises and wrong matching pixels in cost aggregation. These goals are achieved by analyzing the normal curvature of the image surface.

Recently, Xu et al. (2020) proposed an MVS method that considers the normal curvature information for feature extraction. This method handles ambiguity in the patch-match stereo method (Barnes et al., 2009). It selects a patch scale for each pixel by thresholding the normal curvatures computed in several candidate scales. Although the method effectively improves the quality of patch matching, the use of its handcrafted features limits the overall MVS performance. In contrast, our method does not require extracting handcrafted features and determining the threshold for scale selection. Furthermore, CDSConv operates not only on the color information of images but also on the high dimensional features. Multiple CDSConv operations can be stacked to form a very deep architecture for robust feature extraction in MVS.

**Multi-scale features.** Due to large-scale variations of objects and textures in complex scenes, multi-scale representations are adapted for accurate and effective dense pixel-level prediction in many computer vision tasks. For the MVS task, Feature Pyramid Network (FPN) (Lin et al., 2017; Gu et al., 2020; Cheng et al., 2020; Zhang et al., 2020; Luo et al., 2020) is a common approach for multi-scale feature extraction. It is designed as the UNet-liked architecture (Ronneberger et al., 2015) to fuse features from different scales into a single one. However, the multi-scale of FPN is achieved at an image level through down-upsampling operations. It cannot capture the large-scale variation of objects at a pixel level. Several studies in image segmentation can learn pixel-level multi-scale features by using multiple kernels with different sizes (He et al., 2019; Liu et al., 2016; Chen et al., 2017; Zhao et al., 2017). However, these methods prefer a large receptive field for segmentation tasks and usually ignore the details of objects essential for MVS tasks. In contrast to the aforementioned methods (Gu et al., 2020; Cheng et al., 2020; Zhang et al., 2020; Luo et al., 2020; He et al., 2019), our proposed CDSConv can select an optimal scale for each pixel to produce a robust representation and enhance the matching uncertainty. Moreover, our feature extraction CDSFNet can cover denser scales by utilizing the power of a deep CNN.

**Dynamic filtering.** Instead of observing static receptive fields, several studies modified standard convolution to choose the adaptive receptive field for each pixel (Wu et al., 2018; Jia et al., 2016; Han et al., 2018). SAC (Zhang et al., 2017) was proposed to modify the fixed-size receptive field

by learning position-adaptive scale coefficients. Deformable ConvNets (Dai et al., 2017; Zhu et al., 2019) learned offsets for each pixel in a regular sampling grid of standard convolution to enlarge the sampling field with an arbitrary form that can discover geometric-invariant features. Recently, Chen et al. (2020b) proposed a dynamic convolution composed of multiple kernels with the same size to increase model complexity without increasing the network depth or width. All these existing works learn dynamic filters directly from the input images. They are suitable for segmentation or recognition tasks. For MVS task, they ignore the epipolar constraint between reference and source images. In contrast, our dynamic filtering is guided by normal curvature computed in the direction of epipolar line, which can measure the matching capacity between reference and source image patches. To the best of our knowledge, this is the first work that introduces dynamic filters to MVS.

## 3  PROPOSED FEATURE EXTRACTION NETWORK

### 3.1  NORMAL CURVATURE

Normal curvature is used to estimate how much a surface is curved at a specific point along a particular direction (Do Carmo, 2016). Let $p(X, \sigma)$ be a patch centered at pixel $X$ and having a scale $\sigma$ which is proportional to the window size. $L(X, I^{src})$ is denoted as the epipolar line on the reference image $I^{ref}$, which passes through $X$ and is related to a source image $I^{src}$. To reduce the matching ambiguity between $p$ in $I^{ref}$ and the correct patch $p'$ in $I^{src}$, a proper scale $\sigma$ should be chosen when the corresponding patch $p(X, \sigma)$ is less linearly correlated to its surrounding patches along the epipolar line $L$. From the scale space theory (Lindeberg, 1994), the patch $p(X, \sigma)$ can be considered as a point on the image surface represented at the scale $\sigma$. Therefore, the scale $\sigma$ is optimal for stereo matching if the image surface represented in that scale is curved significantly at the point $X$ along the epipolar line direction (Xu et al., 2020). This requires to compute the normal curvature for the patch $p(X, \sigma)$ along the direction $\omega = [u, v]^T$ of the epipolar line $L$. The formula can be expressed in terms of the values of the first and second fundamental forms of the image surface:

$$curv_\sigma(X, \omega) = \frac{FM_{II}}{FM_I} = \frac{1}{\sqrt{1 + I_x^2 + I_y^2}} \frac{\omega \begin{bmatrix} I_{xx} & I_{xy} \\ I_{xy} & I_{yy} \end{bmatrix} \omega^T}{\omega \begin{bmatrix} 1 + I_x^2 & I_x I_y \\ I_x I_y & 1 + I_y^2 \end{bmatrix} \omega^T}, \tag{1}$$

where $I_x, I_y, I_{xx}, I_{xy}, I_{yy}$ are the first-order and second-order derivatives of the image $I$ along the *x-axis* and *y-axis*, respectively.

### 3.2  CURVATURE-GUIDED DYNAMIC SCALE CONVOLUTION

This section describes the novel convolutional module CDSConv to extract dynamic scale features. Given a set of $K$ convolutional kernels with different size $\{C_1, C_2, \ldots, C_K\}$ corresponding to K candidate scales $\{\sigma_1, \sigma_2, \ldots, \sigma_K\}$, CDSConv aims to select a proper scale for each pixel $X$. Hence, it can produce a robust output feature $F_{out}$ from the input $F_{in}$ based on the selected scale. Fig. 1b shows our overall pipeline of CDSConv including two steps. First, CDSConv estimates approximately normal curvatures at $K$ candidate scales. Second, it performs a scale selection step to output the optimal scale from $K$ estimated curvatures. This selection is implemented by a classification network composed of two convolutional blocks and a Softmax activation for the output.

**Learnable normal curvature.** The formula of normal curvature of a patch centered at $X$ along the direction of epipolar line $\omega = [u, v]^T$ in Eq. 1 can be re-written as follows

$$curv_\sigma(X, \omega) = \frac{u^2 I_{xx}(X, \sigma) + 2uv I_{xy}(X, \sigma) + v^2 I_{yy}(X, \sigma)}{\sqrt{1 + I_x^2(X, \sigma) + I_y^2(X, \sigma) \left(1 + (uI_x(X, \sigma) + vI_y(X, \sigma))^2\right)}} \tag{2}$$

where $I(X, \sigma) = I(X) * G(X, \sigma)$ is the image intensity of pixel $X$ in the image scale $\sigma$; it is determined by convolving $I$ with a Gaussian kernel $G(X, \sigma)$ with the window size/scale $\sigma$ (Lindeberg, 1994). The derivatives $I_x, I_y, I_{xx}, I_{xy}$, and $I_{yy}$ can be computed by convolution between the original image $I$ and the derivatives of Gaussian kernel $G(X, \sigma)$

$$\frac{\partial^{i+j}}{\partial x^i \partial y^j} I(X, \sigma) = I(X) * \frac{\partial^{i+j}}{\partial x^i \partial y^j} G(X, \sigma) \tag{3}$$

where $*$ is the convolution operator. There are two main drawbacks when embedding the normal curvature in Eq. 2 into a deep neural network. First, the computation is heavy because of five convolution operations for computing the derivatives $I_x$, $I_y$, $I_{xx}$, $I_{xy}$, and $I_{yy}$. Second, using Eq. 2 to compute curvature is infeasible when the pixel $X$ is a latent feature $F^{in}(X)$ instead of the image intensity $I(X)$. For these reasons, we derive an approximate form of the normal curvature, which reduces the computational cost and can handle the high-dimensional feature input.

To address the heavy computation issue, we notice that the curvature $curv_\sigma$ and derivatives $I_{x^i y^j}$ are proportional to Gaussian kernel $G$, following Eq. 2 and Eq. 3. Moreover, we use a classification network to select the patch scale automatically from the curvature inputs. Therefore, we can perform normalization for the curvatures by rescaling the kernel $G$. We restrict the gaussian kernel in a small range, i.e., $G(X, \sigma) \ll 1$. As a result, the derivatives $I_x$, $I_y$, $I_{xx}$, $I_{xy}$, and $I_{yy}$ are also much smaller than 1. We can approximate the denominator in Eq. 2 by 1. Eq. 2 can be re-defined as

$$curv_\sigma(X, \omega) \approx u^2 I_{xx}(X, \sigma) + 2uv I_{xy}(X, \sigma) + v^2 I_{yy}(X, \sigma) \tag{4}$$

where $curv_\sigma(X, \omega)$, $I_{xx}$, $I_{xy}$ and $I_{yy}$ is much smaller than 1, which implies that their values should be restricted in a small range. The computation in Eq. 4 is reduced by haft compared to Eq. 2.

To make Eq. 4 work with the high-dimensional image feature $F^{in}(X, \sigma)$, we propose to use learnable kernels instead of using fixed derivatives of Gaussian kernel. In particular, for each scale $\sigma$, we introduce three learnable convolution kernels $K_\sigma^{xx}$, $K_\sigma^{xy}$, $K_\sigma^{yy}$ to replace $G_{xx}$, $G_{xy}$, and $G_{yy}$ respectively. These kernels adapt to the input features to approximately compute the second-order derivatives of the image surface. They are trained implicitly by backpropagation when training the end-to-end networks. Weights of these kernels need to be restricted to a small value $K_\sigma^{(\cdot)} \ll 1$ because of the assumption about the Gaussian kernel. We enforce this constraint by adding a regularization term to the loss in Section 4.2.

In summary, we propose learnable normal curvature having the formula as follows (written in the matrix form)

$$curv_\sigma(X, \omega) = \omega \begin{bmatrix} F^{in} * K_\sigma^{xx} & F^{in} * K_\sigma^{xy} \\ F^{in} * K_\sigma^{xy} & F^{in} * K_\sigma^{yy} \end{bmatrix} \omega^T \tag{5}$$

where $K_\sigma^{(\cdot)}$s are the learnable kernels and $\|K_\sigma^{(\cdot)}\| \to 0$, $F^{in}$ is the input feature.

**Scale selection.** After obtaining $K$ normal curvatures $\{curv_{\sigma_1}, curv_{\sigma_2}, \ldots, curv_{\sigma_K}\}$ in $K$ candidate scales, this step selects a proper scale $\sigma_{(\cdot)}$ for each pixel $X$ and then outputs the feature $F^{out}(X)$ from the corresponding convolution kernel $C_{(\cdot)}$. The proper scale is selected by analyzing the curvatures estimated in $K$ candidate scales. It is observed that a small scale cannot capture enough context information to learn discriminative feature, especially in low-textured regions. Meanwhile, a large scale smooths local structure in rich texture regions. MARMVS (Xu et al., 2020) chose the proper scale by searching the normal curvature with a threshold $T = 0.01$. However, using of fixed threshold $T$ prevents generalization on various scenes and structures. Instead, we propose a classification strategy that automatically selects the proper scale from $K$ curvature inputs. This is achieved by using a lightweight CNN with two convolutional blocks. For each pixel $X$, the CNN outputs one-hot vector $[w_1, w_2, \ldots, w_K]$ for scale selection by applying a Softmax with small temperature parameter, which is a differentiable approximation to argmax operator (Jang et al., 2017).

Finally, the feature output $F^{out}$ is produced from the feature input $F^{in}$ with $K$ candidate kernels $\{C_1, C_2, \ldots, C_K\}$ by using weighted sum

$$F^{out} = w_1(F^{in} * C_1) + w_2(F^{in} * C_2) + \ldots + w_K(F^{in} * C_K) \tag{6}$$

where $*$ is the convolution operator. Also, we can extract the normal curvature corresponding to the selected scale by

$$NC^{est} = w_1 curv_{\sigma_1} + w_2 curv_{\sigma_2} + \ldots + w_K curv_{\sigma_K} \tag{7}$$

## 3.3 CURVATURE-GUIDED DYNAMIC SCALE FEATURE NETWORK

This section introduces the Curvature-guided dynamic scale feature network, CDSFNet, which is used as feature extraction step for our MVS framework. CDSFNet is composed of multiple CD-SConv layers instead of standard convolution layers. By expanding the searching scale-space,

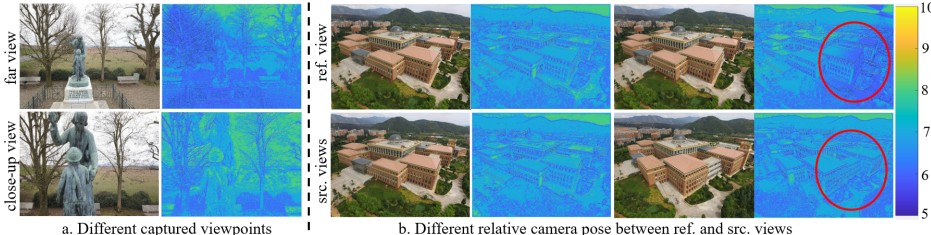

Figure 2: Dynamic scale according to different captured viewpoints and relative camera pose between reference and source views. Figure a shows the scale maps when images is captured at the far view and close-up view. Figure b shows the results when the difference of camera poses is small (left) and large (right).

CDSFNet can select the optimal scale for each pixel to learn robust representation that reduces matching ambiguity.

**Architecture.** CDSFNet is an Unet-liked architecture (Ronneberger et al., 2015). It is organized into three levels of spatial resolution $l \in \{0, 1, 2\}$ to adapt the coarse-to-fine MVS framework. Given the inputs including image $I$ and its estimated epipole $e$, the network outputs three features for three levels $\{F^{(0)}, F^{(1)}, F^{(2)}\}$ and three estimated normal curvatures $\{NC^{est,(0)}, NC^{est,(1)}, NC^{est,(2)}\}$.

**Understanding CDSFNet.** The main goal of CDSFNet is to select pixel-level scale for robust feature extraction. The scale changes dynamically, depending on the detail of image objects and the epipolar constraint, i.e. the relative camera pose between reference and source images. Therefore, our method is more appropriate for MVS than the other dynamic scale networks (He et al., 2019; Wu et al., 2018; Jia et al., 2016) proposed in image segmentation task.

To have a low computational complexity, the number of candidate scales in CDSConv layer should be small, we only use 2 or 3 candidates. However, the searching scale-space is expanded profoundly when multiple CDSConv layers are stacked in CDSFNet. Fig. 2 show the patch scale or window size for each pixel estimated roughly from the first two layers of CDSFNet. For each reference and source views, we draw the reference and source scale maps respectively. There are two main advantages of CDSFNet shown in Fig. 2. First, the estimated scales in rich texture, thin structure, and near-edge regions are often small while those in untextured regions are large. Second, the scales are changed with respect to viewpoints (Fig. 2a) and relative camera pose between reference and source views (Fig. 2b). In Fig. 2a, the closer viewpoint is, the larger scales are estimated. In Fig. 2b, the reference and source scale maps are similar when the difference of camera poses is not large. Otherwise, when the difference is large, the scale map of reference view is changed to adapt the source view, which was marked in the red circle. This is a superiority of CDSFNet because it estimates scale based on epipolar constraint between reference and source views.

## 4  CDS-MVSNET

In this section, we describe the proposed MVS network referred to as CDS-MVSNet, which predicts a depth map $D^{ref}$ from the reference $I^0$ and $N$ source images $\{I_i\}_{i=1}^N$. We adopted the cascade structure of CasMVSNet (Gu et al., 2020) as the baseline structure of CDS-MVSNet. The network is composed of multiple cascade stages to predict the depth maps in a coarse-to-fine manner. Each stage estimates a depth through three steps: feature extraction, cost volume formulation, cost volume regularization & depth regression.

CDS-MVSNet formulates a 3D cost volume based on the output features of CDSFNet. The features of CDSFNet effectively reduce the matching ambiguity by considering the proper pixel scales. Due to the robustness of our feature extraction and cost formulation, CDS-MVSNet can produce more accurate depths in coarser stages compared to other cascade MVS methods (Gu et al., 2020; Cheng et al., 2020; Yang et al., 2020; Wang et al., 2021). There is even little difference in the accuracy of the depths estimated at half resolution and full resolution. Therefore, differently from CasMVSNet, CDS-MVSNet only computes the depths of half-resolution images using three-step MVS computation and then upsamples the depths to the original resolution in the final stage. The upsampled depths at the original resolution can be refined by a 2D CNN to obtain the final depths. This approach can drastically reduce the computation time and GPU memory requirements.

## 4.1 DETAILS OF ARCHITECTURE

**Robust Feature Extraction.** Given the reference image $I_0$ and $N$ source images $\{I_1, I_2, \ldots, I_N\}$, CDSFNet extracts a robust feature pair $(F_{0,i}, F_i)_{i=1\ldots N}$ for each image pair $(I_0, I_i)_{i=1\ldots N}$. Note that the feature of the reference image $F_{0,i}$ varies according to each source image i due to the guidance of epipolar geometry in normal curvature estimation. This is our superiority compared to the state-of-the-art MVS networks which learn an identical feature for the reference image.

**Cost Volume Formulation.** We first compute two-view matching costs and then aggregate them into a single cost. The two-view matching cost is computed per sampled depth hypothesis to form cost volume (Yao et al., 2018; Gu et al., 2020; Wang et al., 2021). Given a sampled depth hypothesis $d_j (j = 1 \ldots D)$ along with all camera parameters, we warp the feature maps of source image $\{F_i\}_{i=1}^N$ into this depth plane. Let $F_i^w(d_j)$ be the warped feature map of source image $i$ at the depth hypothesis $d_j$. The two-view matching cost $V_i(d_j)$ between the reference image $I_0$ and source image $I_i$ at the depth hypothesis $d_j$ is determined as $V_i(d_j) = \langle F_{0,i}, F_i^w(d_j) \rangle$, where $\langle .,. \rangle$ is the pixel-wise inner product for measuring the similarity of two feature maps.

Next, we perform cost aggregation over the computed two-view cost volumes. Inspired from visibility-awareness for cost aggregation (Zhang et al., 2020; Xu & Tao, 2020c), we employ pixel-wise view weight prediction. However, instead of predicting directly from two-view cost volume (Zhang et al., 2020; Xu & Tao, 2020c; Yi et al., 2020), we additionally utilize the normal curvature maps, $NC^{est}$, estimated by CDSFNet in Eq. 7. The normal curvature can implicitly provide information about level details of surface. For instance, a pixel with small normal curvature indicates that it belongs to a large untextured region. In this case, its feature is unmatchable and needs to be eliminated from cost aggregation to reduce noises for cost regularization step.

In summary, the estimated normal curvature encodes implicitly the matching ability of learned feature. Let $NC_i^{est}$ be the estimated normal curvature of the reference view related to source view $i$, and $H_i$ be the entropy computed from two-view matching cost volume $V_i$. We use a simple 2D CNN denoted as $Vis(.)$ to predict a view weight map $W_i$ from two inputs $NC_i^{est}$ and $H_i$. Finally, the aggregated cost $V$ is computed by weighted mean.

$$H_i = -\sum_{j=1}^{D} V_i(d_j) \log V_i(d_j), \quad W_i = Vis(NC_i^{est}, H_i), \quad V(d_j) = \frac{\sum_{i=1}^{N} W_i V_i(d_j)}{\sum_{i=1}^{N} W_i}$$

**Cost volume regularization & Depth regression.** Following most recent learning-based MVS methods, a 3D CNN is applied to obtain a depth probability volume $P$ from the aggregated matching cost volume $V$. The depth is then regressed from $P$ by $D^{est} = \sum_{j=1}^{D} d_j P(d_j)$

## 4.2 LOSS FUNCTION

We propose a feature loss for training CDSFNet effectively. Given a ground truth depth $D^{gt}$ of the reference image, we warp the feature of source images into this depth map and then compute the matching cost $V(D^{gt})$. We label this map as positive, hence defines its classification label as $c = 1$. To generate the matching costs with negative labels $c = 0$, we randomly sample $N_d$ neighboring depths $D^{neg}$ around the ground truth $D^{gt}$ and then compute the matching costs $V(D^{neg})$ for these depths. Finally, binary cross entropy is used to define the feature loss. However, we need to add regularization terms to restrict the convolutional weights of CDSFNet, $w_{CDSF}$, and estimated normal curvature $NC^{est}$ to a small range as mentioned in Section 3.2. Therefore, the final feature loss is defined as:

$$\mathcal{L}_{feat} = \frac{1}{M} \sum_{X} \log \left( \text{sig}(V(D^{gt}))(1 - \text{sig}(V(D^{neg}(X)))) \right) + \lambda_1 \|w_{CDSF}\|^2 + \frac{\lambda_2}{M} \|NC^{est}\|^2 \quad (8)$$

where sig$(.)$ is the sigmoid function, $M$ is total number of pixels $X$, $\lambda_1 = 0.01$ and $\lambda_2 = 0.1$ are the regularized hyperparameters.

Similar to most learning-based MVS methods (Yao et al., 2018; Gu et al., 2020), the depth loss $\mathcal{L}_{depth}$ is defined by $L1$ loss between the predicted depth and the ground truth depth. Because our method includes 4 stages, there are four depth loss $\{\mathcal{L}_{depth}^{(l)}\}_{l=0}^3$. Finally, the total loss $\mathcal{L}_{total}$ is defined as a sum of two mentioned losses $\mathcal{L}_{total} = 5\mathcal{L}_{feat} + \sum_{l=0}^{3} \mathcal{L}_{depth}^{(l)}$

Table 1: Quantitative results on the DTU evaluation dataset (lower is better).

| Methods | Mean error distance (mm) | | |
|---|---|---|---|
| | Acc. | Comp. | Overall |
| **Gipuma**[*] (Galliani et al., 2015) | **0.283** | 0.873 | 0.578 |
| **Colmap**[*] (Schönberger et al., 2016) | 0.400 | 0.664 | 0.532 |
| **P-MVSNet**[°] (Luo et al., 2019) | 0.406 | 0.434 | 0.420 |
| **Fast-MVSNet**[°] (Yu & Gao, 2020) | 0.336 | 0.403 | 0.370 |
| **Vis-MVSNet**[°] (Zhang et al., 2020) | 0.369 | 0.361 | 0.365 |
| **AttMVS**[†] (Luo et al., 2020) | 0.383 | 0.329 | 0.356 |
| **CasMVSNet**[°] (Gu et al., 2020) | 0.325 | 0.385 | 0.355 |
| **UCSNet**[°] (Cheng et al., 2020) | 0.338 | 0.349 | 0.344 |
| **PatchMatchNet**[°] (Wang et al., 2021) | 0.427 | **0.277** | 0.352 |
| **PVA-MVSNet**[°] (Yi et al., 2020) | 0.379 | 0.336 | 0.357 |
| **CVP-MVSNet**[°] (Yang et al., 2020) | 0.296 | 0.406 | 0.351 |
| **BP-MVSNet**[†] (Sormann et al., 2020) | 0.333 | 0.320 | 0.327 |
| **Ours**[°] (DTU only) | 0.352 | **0.280** | **0.316** |
| **Ours** (DTU+BlendedMVS) | 0.351 | **0.278** | **0.315** |

[*] traditional method, [°] trained on training set of DTU, [†] trained on a modified version of DTU

Table 2: Evaluation of estimated depth with different image resolutions on DTU (higher is better).

| | Methods | Resolutions (width x height) | | | | |
|---|---|---|---|---|---|---|
| | | 832 x 576 | 1152 x 832 | 1280 x 960 | 1408 x 1088 | 1536 x 1152 |
| Prec. 2mm (%) | CasMVSNet | **76.54** | **75.35** | 74.01 | 72.22 | 70.92 |
| | PatchMatchNet | 69.52 | 71.23 | 71.37 | 71.20 | 70.99 |
| | UCSNet | 74.11 | 73.83 | 72.95 | 71.58 | 70.49 |
| | CVP-MVSNet | 66.80 | 69.75 | 70.71 | 71.03 | 71.16 |
| | Ours | 72.71 | 75.04 | **75.09** | **74.91** | **74.64** |
| Prec. 4mm (%) | CasMVSNet | 82.11 | 80.47 | 79.02 | 77.32 | 76.15 |
| | PatchMatchNet | 78.31 | 78.32 | 78.03 | 77.67 | 77.34 |
| | UCSNet | **82.18** | 80.10 | 78.59 | 76.91 | 75.61 |
| | CVP-MVSNet | 75.28 | 77.41 | 78.18 | 78.68 | 78.76 |
| | Ours | 80.86 | **81.38** | **80.99** | **80.48** | **80.02** |

# 5 EXPERIMENTAL RESULTS

## 5.1 DATASETS

The datasets used in our evaluation are DTU (Aanæs et al., 2016), BlendedMVS (Yao et al., 2020), and Tanks & Temples (Knapitsch et al., 2017). Due to the simple camera trajectory of all scenes in DTU, we additionally utilize the BlendedMVS dataset with diverse camera trajectories for training. Tanks & Temples is provided as a set of video sequences in realistic environments. We train the model on the training sets of DTU and BlendedMVS, then evaluate the testing set of DTU and intermediate set of Tanks & Temples. Our source code is available at `https://github.com/TruongKhang/cds-mvsnet`.

## 5.2 BENCHMARK PERFORMANCE

**Results for DTU dataset.** We predict the depth at the highest resolution $1536 \times 1152$. The performance of the proposed method was compared with state-of-the-art MVS methods, including conventional and learning-based methods, using the DTU dataset. However, we especially notice the learning-based methods including CasMVSNet, UCSNet, CVP-MVSNet, and Vis-MVSNet, which adopted the cascade structure for MVS depth estimation. Similar to us, these methods used the same 3D CNN cost regularization. Therefore, we can validate our feature extraction and cost formulation by comparing with them.

Table 1 presents the quantitative results achieved by the methods on the DTU evaluation dataset. We calculated the three standard error metrics (Aanæs et al., 2016) given on the official site of DTU: accuracy, completeness, and overall error. Following the same setups with most of the baselines, we trained our model on the DTU training set and then used this model for evaluation. Moreover, we also provided the results which produced by training on both DTU and BlendedMVS simultaneously. In both cases, our method exhibited the best performance in terms of completeness and overall error. We also achieved good results for both accuracy and completeness, while the other methods often produced a large trade-off between these two metrics. For example, Gipuma had a high accuracy at 0.283 but it produced very low completeness, 0.873. In contrast, PatchMatchNet was with a high completeness (0.277) and very low accuracy (0.427). Moreover, our method outperformed all similar methods using the cascade structure. This indicates that our feature representation and cost aggregation guided by normal curvature can remarkably improve MVS depth prediction remarkably.

To validate the dynamic scale property of our CDSFNet, we measure depth precision on different image resolutions. Depth precision is the average percentage of depths with errors lower than a defined threshold (Gu et al., 2020). Here we computed precision with the thresholds of 2mm and 4mm. We compared to CasMVSNet, PatchMatchNet, UCSNet, and CVP-MVSNet, which applied the cascade structure to predict high-resolution depths. Note that all methods are trained on the datasets with image-resolution of $640 \times 512$. Table 2 presents the quantitative results for the depth estimated on five image resolutions. The baseline methods degraded the performance when the image-resolution was increased while our method could maintain a stable performance at any image resolution. The reason is that our feature extraction CDSFNet can utilize dynamic scales for learning features. The dynamic scale adapts to various image-resolutions and object scales in image.

Table 3: Performance comparisons (F-score) of various reconstruction algorithms on the intermediate sequences of the Tanks & Temples benchmark. The higher value of F-score implies better reconstruction results. Our method achieves the best performance in terms of mean F-score.

| Method | Mean | Family | Francis | Horse | Lighthouse | M60 | Panther | Playground | Train |
|---|---|---|---|---|---|---|---|---|---|
| ACMM* (Xu & Tao, 2019) | 57.27 | 69.24 | 51.45 | 46.97 | 63.20 | 55.07 | 57.64 | 60.08 | 54.48 |
| ACMP* (Xu & Tao, 2020b) | 58.41 | 70.30 | 54.06 | 54.11 | 61.65 | 54.16 | 57.60 | 58.12 | 57.25 |
| Fast-MVSNet° (Yu & Gao, 2020) | 47.39 | 65.18 | 39.59 | 34.98 | 47.81 | 49.16 | 46.20 | 53.27 | 42.91 |
| Vis-MVSNet[†] (Zhang et al., 2020) | 60.03 | 77.40 | 60.23 | 47.07 | 63.44 | 62.21 | 57.28 | 60.54 | 52.07 |
| AttMVS[◊] (Luo et al., 2020) | 60.05 | 73.90 | 62.58 | 44.08 | 64.88 | 56.08 | 59.39 | 63.42 | 56.06 |
| CasMVSNet° (Gu et al., 2020) | 56.84 | 76.37 | 58.45 | 46.26 | 55.81 | 56.11 | 54.06 | 58.18 | 49.51 |
| UCSNet° (Cheng et al., 2020) | 54.83 | 76.09 | 53.16 | 43.03 | 54.00 | 55.60 | 51.49 | 57.38 | 47.89 |
| PVA-MVSNet° (Yi et al., 2020) | 54.46 | 69.36 | 46.80 | 46.01 | 55.74 | 57.23 | 54.75 | 56.70 | 49.06 |
| CVP-MVSNet° (Yang et al., 2020) | 54.03 | 76.50 | 47.74 | 36.34 | 55.12 | 57.28 | 54.28 | 57.43 | 47.54 |
| BP-MVSNet[†] (Sormann et al., 2020) | 57.60 | 77.31 | 60.90 | 47.89 | 58.26 | 56.00 | 51.54 | 58.47 | 50.41 |
| Ours[†] (fine-tuning on BlendedMVS) | 60.82 | 78.17 | 61.74 | 53.12 | 60.25 | 61.91 | 58.45 | 62.35 | 50.58 |
| Ours (DTU+BlendedMVS) | 61.58 | 78.85 | 63.17 | 53.04 | 61.34 | 62.63 | 59.06 | 62.28 | 52.30 |

⋆ traditional method, ° only training on the training set of DTU, † training on DTU and then fine-tuning on BlendedMVS, ◊ training on a modified version of DTU

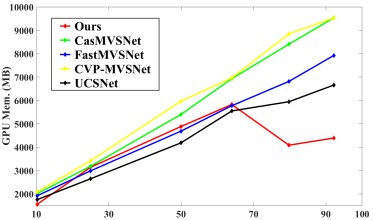
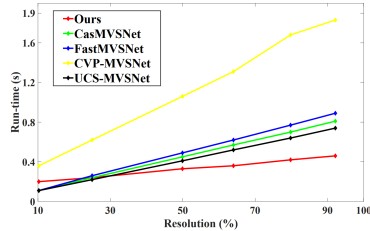

Figure 3: Relating GPU memory and run-time to the input resolution on DTU's evaluation set. The original image resolution is $1600 \times 1200$ (100%). Here we evaluate at the highest resolution $1536 \times 1152$ (95.2%).

**Generalization on Tanks & Temples Dataset.** To verify the generalization capability of our CDS-MVSNet, we also provided two evaluations similar to the evaluation of DTU. First, we fine-tuned the pre-trained model on DTU by using BlendedMS dataset. Second, we used directly the model trained on both datasets for evaluation. The reconstructed point clouds were submitted to the benchmark website of Tanks & Temples (Knapitsch et al., 2017) to receive the F-score, which is a combination of precision and recall of the reconstructed 3D model. Table 3 lists the quantitative results of our method and other state-of-the-art methods. As shown in Table 3, our method achieved the best mean Fscores for both evaluations, 60.82 and 61.58, respectively. Our method outperformed both the state-of-the-art traditional methods such as ACMM, ACMP and the learning-based methods including AttMVS and Vis-MVSNet. The qualitative results of all scenes are shown in detail in the supplemental materials. Compared to CasMVSNet and UCSNet, which are most similar to our method except the feature extraction and cost formulation, we obtained better mean F-score with high margins. This demonstrates the effectiveness of our feature extraction CDSFNet and cost aggregation strategy on complex outdoor scenes.

**Run-time and Memory.** This section evaluates the memory consumption and run-time compared to several state-of-the-art learning-based methods that achieve competing performance and employ the cascade structure similar to us: CasMVSNet, UCS-Net, CVP-MVSNet, and FastMVSNet. Fig. 3 shows the memory and run-time of all methods measured on DTU evaluation set with different image resolutions. We observe that the run-time and memory of all methods are more required when the image-resolution increases higher. However, we notice that the memory consumption of our method was dropped from the resolution of 80% while our run-time was slightly increased. This indicates that our method can release a significant amount of GPU memory by only sacrificing a small amount of run-time. Our method achieves significantly faster run-time and lower required memory at the very high image-resolution compared to the other baselines. For example, at a resolution of $1536 \times 1152$ (92.2%), the memory consumption is reduced by **54.0%**, 34.1%, and **54.0%** compared to CasMVSNet, UCSNet, and CVP-MVSNet respectively. The corresponding comparison results of run-time are 43.2%, 37.8%, and **74.9%**. Finally, we conclude that our method is more efficient in memory consumption and run-time than most state-of-the-art learning-based methods.

## 6 CONCLUSION

We propose a dynamic scale network CDSFNet for MVS task. CDSFNet can extract the discriminative features to improve the matching ability by analyzing the normal curvature of the image surface. We then design an efficient MVS architecture, CDS-MVSNet, which achieves state-of-the-art performance on various outdoor scenes

ACKNOWLEDGMENTS

This work was supported by the National Research Foundation of Korea (NRF) funded by the Ministry of Education under Grant 2016R1D1A1B01013573.

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

## A APPENDIX

### A.1 DERIVED FORMULAS OF NORMAL CURVATURE

First, we re-derive Eq. 1 proposed by Xu et al. (2020). The image surface can be represented by

$$s(x, y) = [x, y, I(x, y)],$$

where $X = [x, y]$ is a pixel coordinate and $I(x, y)$ is the color information of image. The unit normal vector of surface $s$ at point $X$ is determined as

$$n = \frac{s_x \wedge s_y}{\|s_x \wedge s_y\|} = \frac{1}{\sqrt{1 + I_x^2 + I_y^2}}[-I_x, -I_y, 1],$$

where $s_x$ and $s_y$ are the first partial derviatives with respect to *x-axis* and *y-axis*, $\wedge$ is the cross-product operator. Then, the first and second fundamental form of surface $s$ at point $X$ can be written by

$$F_I = \begin{bmatrix} s_x.s_x & s_x.s_y \\ s_x.s_y & s_y.s_y \end{bmatrix} = \begin{bmatrix} 1 + I_x^2 & I_x I_y \\ I_x I_y & 1 + I_y^2 \end{bmatrix}$$

$$F_{II} = \begin{bmatrix} s_{xx}.n & s_{xy}.n \\ s_{xy}.n & s_{yy}.n \end{bmatrix} = \frac{1}{\sqrt{1 + I_x^2 + I_y^2}} \begin{bmatrix} I_{xx} & I_{xy} \\ I_{xy} & I_{yy} \end{bmatrix}$$

Note that "·" is the dot-product operation between two vectors. Finally, the normal curvature of $s$ at point $X$ along the direction $\omega = [u, v]^T$ of epipolar line is defined as

$$curv_\sigma(X, \omega) = \frac{FM_{II}}{FM_I} = \frac{1}{\sqrt{1 + I_x^2 + I_y^2}} \frac{\omega \begin{bmatrix} I_{xx} & I_{xy} \\ I_{xy} & I_{yy} \end{bmatrix} \omega^T}{\omega \begin{bmatrix} 1 + I_x^2 & I_x I_y \\ I_x I_y & 1 + I_y^2 \end{bmatrix} \omega^T}$$

$$= \frac{1}{\sqrt{1 + I_x^2 + I_y^2}} \frac{u^2 I_{xx} + 2uv I_{xy} + v^2 I_{yy}}{u^2(1 + I_x^2) + 2uv I_x I_y + v^2(1 + I_y^2)}$$

$$= \frac{1}{\sqrt{1 + I_x^2 + I_y^2}} \frac{u^2 I_{xx} + 2uv I_{xy} + v^2 I_{yy}}{u^2 + v^2 + (uI_x + vI_y)^2}$$

Because $\omega = [u, v]^T$ is an unit vector, $u^2 + v^2 = 1$. Therefore,

$$curv_\sigma(X, \omega) = \frac{u^2 I_{xx} + 2uv I_{xy} + v^2 I_{yy}}{\sqrt{1 + I_x^2 + I_y^2} \left(1 + (uI_x + vI_y)^2\right)}$$

This formula can be applied to an image patch $p$ with a scale $\sigma$ centered at $X$. Following the scale space theory (Lindeberg, 1994), the patch p can be regarded as a pixel when the image I is represented at the scale $\sigma$. Let $I(X, \sigma)$ be the image intensity at the scale $\sigma$. $I(X, \sigma)$ can be computed from $I(X)$ by using a Gaussian kernel with size $\sigma$, $I(X, \sigma) = I(X) * G(X, \sigma)$

Therefore, we can finally write the normal curvature computed for $X$ at the scale $\sigma$ as

$$curv_\sigma(X, \omega) = \frac{u^2 I_{xx}(X, \sigma) + 2uv I_{xy}(X, \sigma) + v^2 I_{yy}(X, \sigma)}{\sqrt{1 + I_x^2(X, \sigma) + I_y^2(X, \sigma)} \left(1 + (uI_x(X, \sigma) + vI_y(X, \sigma))^2\right)},$$

which was Eq. 2 mentioned in Section 3.2

## A.2  DETAIL OF MVS ARCHITECTURE

Fig. 4 depicts the multi-stage architecture of CDS-MVSNet. First, CDS-MVSNet uses the feature extraction CDSFNet to extract outputs at three resolutions $\{\frac{1}{8}, \frac{1}{4}, \frac{1}{2}\}$ corresponding to the first three stages $\{0, 1, 2\}$. For each of these stages $l \in \{0, 1, 2\}$, CDS-MVSNet predicts the depth at the corresponding image-resolution $\frac{W}{2^{3-l}} \times \frac{H}{2^{3-l}}$ by using three-step MVS as described in section 4.1. CDS-MVSNet also uses an independent 3D CNN for each stage to regularize 3D cost volume. The estimated depth of current stage is upsampled and used to reduce the depth hypothesis range of 3D cost volume in the next stage. For the last stage $l = 3$, CDS-MVSNet outputs the full-resolution depths by performing depth refinement from the upsampled depth and the color image.

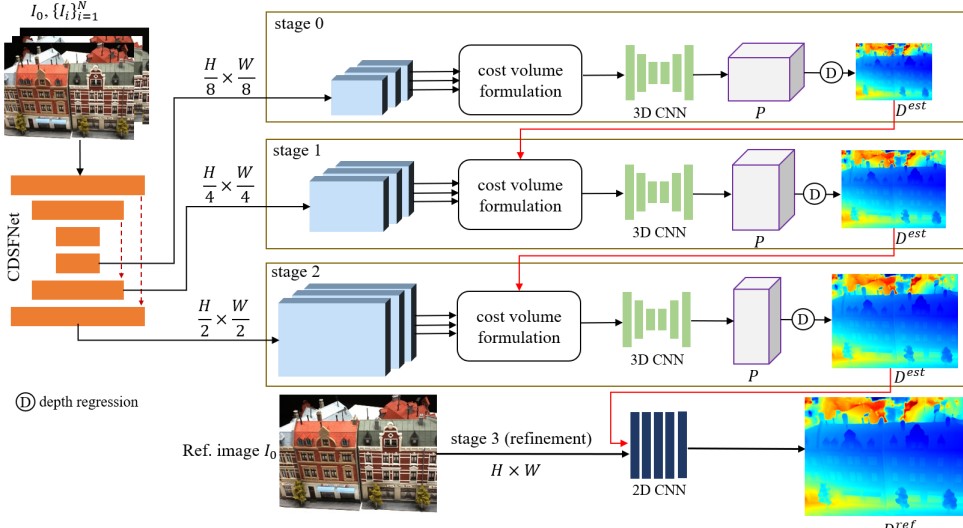

Figure 4: CDS-MVSNet architecture. For the first three stages $l \in \{0, 1, 2\}$, the estimated depth of current stage is used to reduce the depth hypothesis range of a cost volume in the next stage. For the last stage $l = 3$, we perform depth refinement from the estimated depth of previous stage and the color information

**CDSFNet** Given a reference image $I_0$ and $N$ source images $\{I_1, I_2, \ldots, I_N\}$ with corresponding camera parameters $\{Q_0, \ldots, Q_N\}$, we can compute an epipole pair $\{(e_{0,i}, e_i)\}_{i=1}^N$ for each reference-source image pair $\{(I_0, I_i)\}_{i=1}^N$ based on the epipolar geometry. We use the epipole to determine the epipolar line for each pixel. Therefore, given the inputs including image $I$ and epipole $e$, CDSFNet extracts three feature maps corresponding to the three levels $\{0, 1, 2\}$ of CDS-MVSNet. Fig. 5 describes the Unet-liked architecture of CDSFNet. Fig. 6 and 7 show the results of dynamic scale estimation and normal curvature estimation respectively.

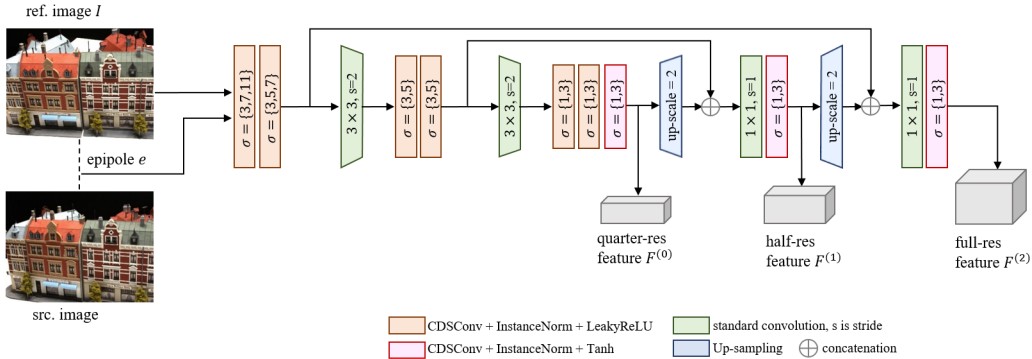

Figure 5: The Unet-liked architecture of CDSFNet. It is organized into three levels: quater-resolution, haft-resolution, and full-resolution

**Depth Refinement.** Instead of using three-step MVS depth estimation as described for the finest resolution level (stage 3), we directly up-sample the depth estimated in previous stage (from resolution $\frac{W}{2} \times \frac{H}{2}$ to $W \times H$) and refine the upsampled depth with the RGB image. Following MSG-Net (Hui et al., 2016) and PatchMatchNet (Wang et al., 2021), we implement a depth residual network to perform refinement. The network uses a 2D CNN to output a residual from the upsampled depth. This residual depth is then added to the up-sampled depth, to get the refined depth map $D^{ref}$.

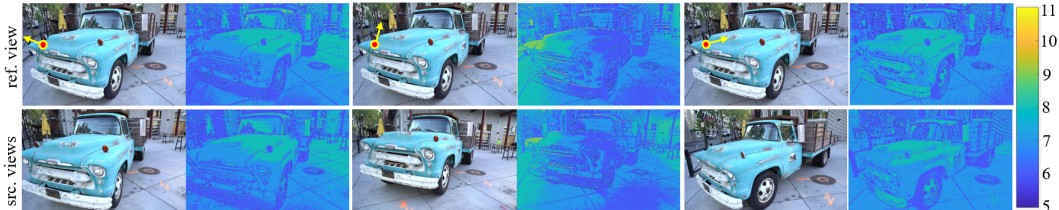

Figure 6: Illustrations for dynamic scale estimation according to epipolar constraint between reference and source views. Consider a pixel of reference view marked in red color, its epipolar line, marked in yellow, varies with each source view. Therefore, the normal curvatures estimated at that pixel are different, this leads to the different estimated scale in the reference view when it is matched to a source view.

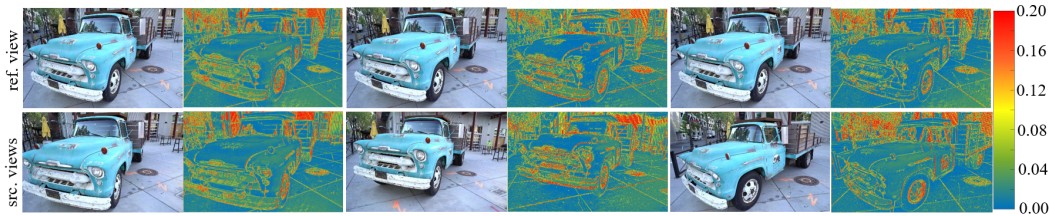

Figure 7: Visualization of the normal curvature maps estimated from our feature extraction CDSFNet by Eq. 7. For each pixel $X$, the normal curvature computed at $X$ is large when $X$ belongs to a rich texture region. In this case, the estimated scale for $X$ is small as shown in Fig. 6 to preserve the local detail of the scenes. Otherwise, the curvature is small when $X$ belongs to an untextured region; therefore, a large scale is chosen to increase the completeness. These figures demonstrate that the normal curvature can provide guidance to choose the optimal scale to improve the matching ability.

### A.3 IMPLEMENTATION DETAILS

**Training.** We applied the same image-resolution $640 \times 512$ in both DTU and BlendedMVS datasets. The preprocessing of these datasets for view selection strategy and ground-truth depth maps was provided by Yao et al. (2018). For CDS-MVSNet, we set the number of input views to 3, the number of depth hypothesis planes to $\{48, 32, 8\}$ with the corresponding depth interval scales $\{4, 2, 1\}$. To train CDFSNet effectively, we used the feature loss as mentioned in Section 4.2. Besides, we implemented the scale selection step of CDSConv by first initializing the Softmax temperature by 1. This hyperparameter was then decreased over training time until it reached $0.01$ to produce the sparse output for scale selection. The entire network CDS-MVSNet was trained with 30 epochs, using the SGD optimizer with an initial learning rate of $0.01$, and on two NVIDIA Titan V GPU with a batch size of 6. We provided two experimental scenarios. First, we trained the model on DTU training set and then evaluated on DTU test set. This pre-trained model is then fine-tuned on BlendedMVS dataset for 15 epochs with a learning rate of $10^{-4}$ to evaluate on Tanks & Temples. Second, we trained a single model on both DTU and BlendedMVS datasets and use this model for evaluation. This training strategy can improve the performance as indicated in Table 1 and 3.

**Point cloud generation.** To reconstruct the final point cloud, we filtered out the low confidence depths based on the probability maps estimated from the depth probability volumes of all stages (Zhang et al., 2020). Similar to (Gu et al., 2020; Zhang et al., 2020), we then integrated all the depth maps based on the fusion method proposed in (Galliani et al., 2015; Yao et al., 2018).

### A.4 ABLATION STUDY

**Effectiveness of learnable curvature.** We demonstrate the advantage of our proposed learnable curvature in comparison to the original curvature (Xu et al., 2020). We replace the curvature estimation in Eq.5 with Eq.1 and then re-train the MVS network on the DTU dataset with identical setups. Table 4 presents four model variants denoted as A, B, C, and D. Each variant implements a specific version of curvature (original or learnable) and then uses the visibility aggregation with or without the curvature prior. We consider a comparison of the two groups, models A and B versus models C

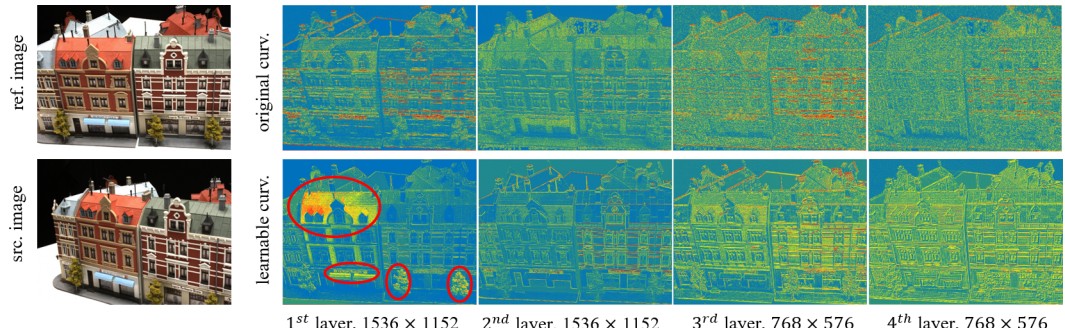

$1^{st}$ layer, $1536 \times 1152$    $2^{nd}$ layer, $1536 \times 1152$    $3^{rd}$ layer, $768 \times 576$    $4^{th}$ layer, $768 \times 576$

Figure 8: A qualitative comparison on DTU evaluation set (Aanæs et al., 2016) between the learnable and original curvature. We visualize the curvature maps at the scale $\sigma = 3$ (kernel size is 3) extracted from the first four layers of CDSFNet. The first and second layers output the curvature maps at the full resolution, the third and fourth layers extract the maps at the half-resolution. The red circles denote the effect of attention mechanism, which is benefited from the learnable kernels $K$ as presented in Section 3.2

Table 4: Comparison between our learnable curvature and the original curvature (Xu et al., 2020). All models are trained on a subset of DTU training set, using the same setups and hyperparameters. The evaluation of depth map and pointcloud is collected on DTU validation and test set, respectively.

| Methods | Design Options | | Depth Map | | | Pointcloud | | | Time & Memory | |
|---------|---------------|----------------|-------------|-------------|-------------|-------------|--------------|-----------------|-------------|-------------|
| | curv. type | vis. with curv. | Prec. 2mm | Prec. 4mm | MAE (mm) | Acc. (mm) | Comp. (mm) | Overall (mm) | Time (s) | Mem (Mb) |
| Model A | original | | 74.26 | 84.89 | 6.84 | 0.378 | 0.315 | 0.347 | 0.539 | 7993 |
| Model B | original | ✓ | 74.12 | 84.77 | 5.99 | 0.391 | 0.327 | 0.359 | 0.539 | 7993 |
| Model C | learnable | | 76.14 | **86.08** | 6.30 | 0.373 | **0.302** | **0.338** | **0.421** | **4389** |
| Model D | learnable | ✓ | **76.23** | 85.92 | **5.89** | **0.372** | 0.305 | 0.339 | **0.421** | **4389** |

and D. To evaluate the depth maps, we measure the percentage depth precision with the thresholds $\{2, 4\}$mm and Mean Absolute Error (MAE). To evaluate the reconstructed models, we use the accuracy, completeness, and overall metrics. We also provide the runtime and memory consumption to evaluate the computational efficiency.

Table 4 indicates that the models with learnable curvature (C and D) achieve a better performance than the original curvature (A and B) for all evaluation metrics. The learnable curvature has a low computation cost and boosts the quality of depth and reconstruction significantly. The original curvature degrades the performance because it cannot handle the high-dimensional features. The original method treats each feature channel equally and averages all feature channels when computing the derivatives by Gaussian filters. Therefore, it may produce a high curvature estimation even though the neighboring feature vectors are visually similar to each other, as shown in Fig. 8.

Fig. 8 shows the normal curvatures estimated by the original method (Xu et al., 2020) and our learnable method. The curvature maps are extracted from the first four CDSConv layers of CDSFNet. As shown in Fig. 8, the original method provides a ground truth estimation only in the first layer when the input is a color image (i.e., the estimated curvatures are small in untextured regions while large in near-edge or rich-textured regions). However, it produces noisy estimation when the inputs are the feature maps in the subsequent layers, especially in the third and fourth layers where the curvatures do not reflect the edge or texture information. On the other hand, although the learnable curvature approximates the ground truth (the original curvature in the first layer), it consistently preserves the properties of normal curvature concerning the untextured and rich-textured regions in every layer. We further notice that our learning-based estimation takes advantage of the attention mechanism, as shown in red circles. It focuses on the rich texture region to improve the completeness of reconstruction, which is illustrated in Fig. 9 in Appendix A.5. Fig. 8 also implies that the estimated curvature tends to be higher at the deeper layers because it encodes a larger patch scale.

**Curvature-guided visibility aggregation.** We proposed a pixel-wise visibility prediction method for cost aggregation in Section 4.1. Previous works generally predicted the visibility weights only from the two-view cost volume (Zhang et al., 2020; Xu & Tao, 2020c). Compared to these works,

Table 5: Impact of the number of candidate scales in CD-SConv layer. The models are trained on DTU training set at the resolution $320 \times 256$, without 3D cost regularization. The statistics below are collected on DTU validation set.

Table 6: The accuracy of estimated depth over different stages in our CDS-MVSNet. The entire MVS network is trained on DTU training set. The trained model is used to evaluate on the DTU test set at the resolution $1536 \times 1152$

| #scales/layer ($N_c$) | Prec. 2mm | Prec. 4mm | Prec. 8mm | MAE (mm) | Time (s) |
|---|---|---|---|---|---|
| 1 (CDSFNet → FPN) | 40.87 | 55.95 | 66.72 | 21.41 | **0.114** |
| 2 | 44.46 | 58.10 | 67.89 | 18.89 | 0.134 |
| 3 | 45.27 | 59.14 | 69.17 | 17.46 | 0.155 |
| 4 | **46.77** | **60.63** | **70.61** | **16.10** | 0.195 |

| Stages | Prec. 1mm | Prec. 2mm | Prec. 4mm | MAE (mm) |
|---|---|---|---|---|
| 0 | 39.99 | 61.89 | 76.35 | 11.43 |
| 1 | 58.01 | 72.92 | 80.21 | 10.80 |
| 2 | 63.97 | 75.43 | 80.92 | 10.67 |
| 3 | **64.26** | **75.52** | **80.94** | **10.62** |

our method additionally considers the curvature prior estimated by CDSFNet. To evaluate the effectiveness of curvature prior to visibility prediction, we compared the performance of our method with the baseline method Zhang et al. (2020). To implement the baseline method, we removed the curvature input from our visibility prediction network and only used the entropy information of two view cost volumes. We compared the models using curvature prior (Models B and D) with those without using it (Models A and C). Table 4 shows the experimental results. As shown in Table 4, Model B had the worst performance in terms of overall. Because Model B produced the noisy curvature, this noisy estimation degraded the performance of Model B compared to Model A. Model D remarkably improves the depth estimation performances remarkably and achieves the similar performance of reconstruction compared to Model C without increasing runtime and memory consumption. These results demonstrate the effectiveness of the proposed learnable curvature and curvature-guided visibility prediction in the MVS process.

**Analysis of CDSFNet and the cascade structure.** This section first analyzes the effectiveness of proposed CDSFNet when increasing the number of candidate scales. To measure explicitly the matchability of features extracted by CDSFNet, we remove the cost-volume regularization step in the MVS architecture and then regress the depth directly from the aggregated 3D cost volume. Table 5 presents the results when increasing the number candidate scales $N_c$ in each CDSConv layer. When $N_c$ is equal to 1, CDSFNet is considered as Feature Pyramid Network (FPN), which is applied to most of the state-of-the-art learning-based methods (Gu et al., 2020; Zhang et al., 2020). As shown in Table 5, CDSFNet achieved better performance when using more candidate scales. However, it suffered from heavy computation. Therefore, we chose the design with two or three scales for each CDSConv (CDSFNet architecture as shown in Fig. 5), to guarantee both the performance of reconstruction and computational efficiency.

Next, we investigate the improvement of depth estimation over the cascade stages. Table 6 provides the accuracy of depth estimation over all stages of our CDS-MVSNet. The first three stages mainly contribute to the final quality of estimated depth. The last stage $l = 3$ is used for depth upsampling; it outputs the full-resolution depth while maintaining the same depth accuracy compared to the previous stage. Suppose this stage is implemented by the three-step MVS depth estimation pipeline mentioned in Section 4.1, the entire MVS network suffers from a considerable computation and cannot fit into a limited GPU memory. Therefore, we only apply the depth refinement with a 2D CNN to the last stage for efficiency in time and memory.

## A.5 QUALITATIVE RESULTS ON DTU AND TANKS & TEMPLES

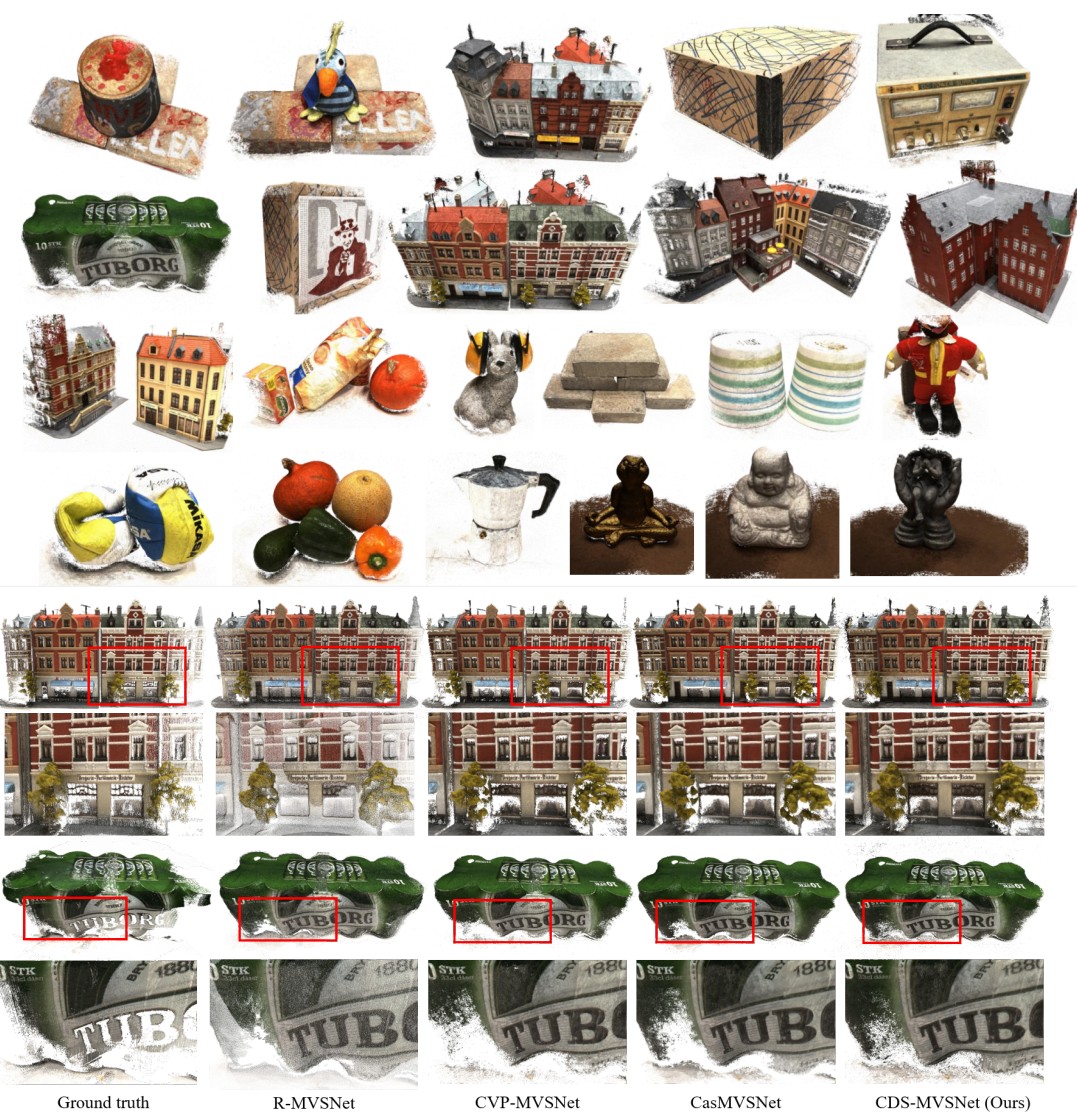

| Ground truth | R-MVSNet | CVP-MVSNet | CasMVSNet | CDS-MVSNet (Ours) |

Figure 9: Qualitative results on DTU evaluation dataset. Our method achieved the high completeness of reconstructed models. The visualization of reconstructed models for DTU evaluation set are shown in the top figure. A comparison with the other state-of-the-art methods is shown in the bottom figure.

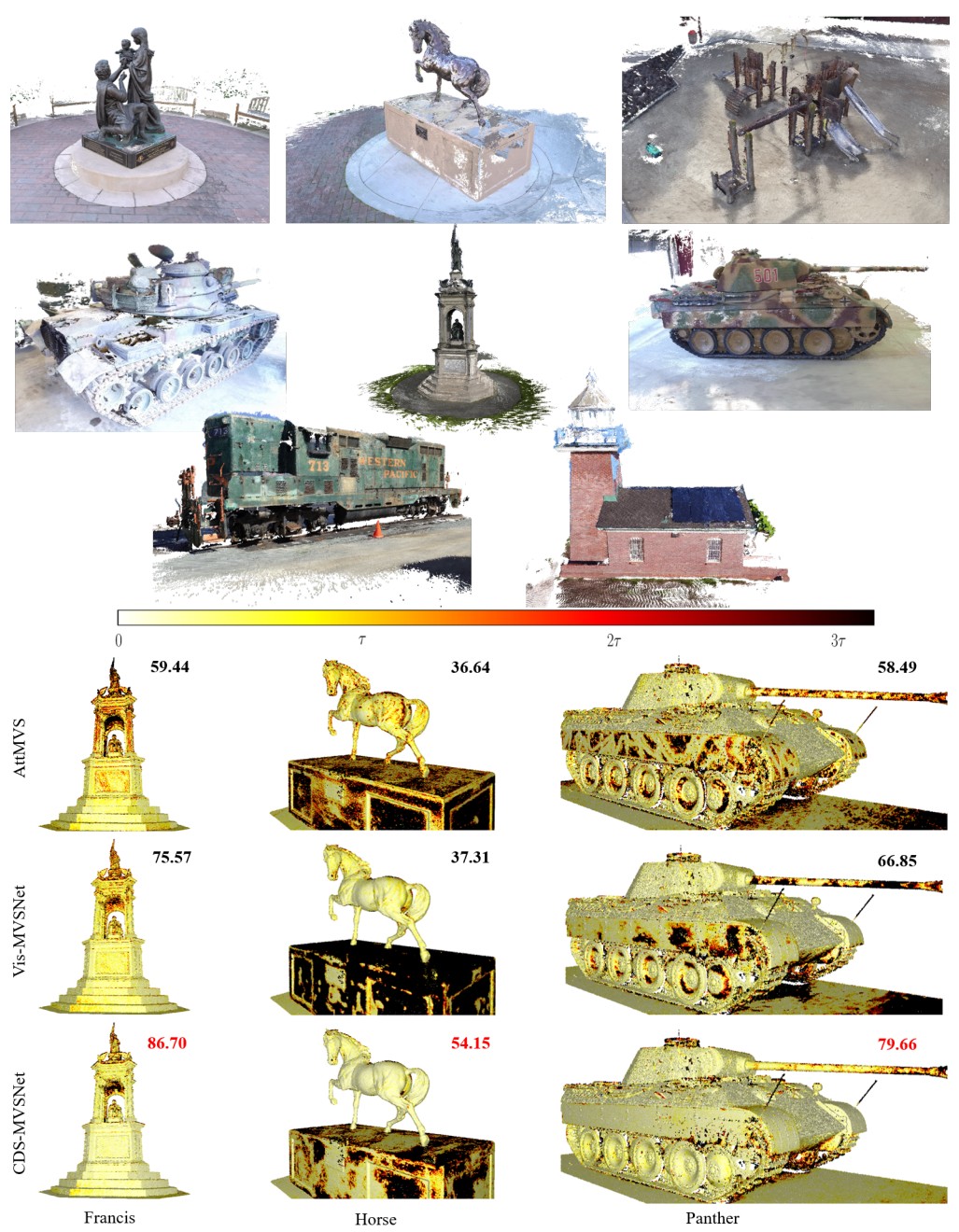

Figure 10: Qualitative results on the intermediate set of Tanks & Temples. Our method achieved the high completeness of reconstruction for complex outdoor scenes. The bottom figure shows the error visualization for Recall scores compared to AttMVS (Luo et al., 2020) and Vis-MVSNet (Zhang et al., 2020)

