# OpenReview forum: "CURVATURE-GUIDED DYNAMIC SCALE NETWORKS FOR MULTI-VIEW  STEREO"
_ICLR.cc/2022/Conference — ICLR 2022 Poster_

### Official Review · Reviewer_6P6T · 2021-10-29

**Correctness:** 3
**Technical Novelty And Significance:** 3
**Empirical Novelty And Significance:** 3
**Recommendation:** 6
**Confidence:** 3

**Main Review:**

The paper is generally well written and the main novelty of the curvature estimation is explained in a high level allowing an inexpert to follow. The authors properly cite previous works when necessary (except Do Carmo which is not the earliest use of curvature AFAIK)

This paper pushes the envelope and seem to provide better results in using less compute, which (IMHO) brings it over the conference's bar, but not by much.

The main area where I believe this paper can be improved is by providing deeper analysis of the novel parts.
Specifically:
- Replacing Gxx with learned Kxx seems arbitrary: first, using G is quite tractable (only 3 derivatives). second, what promise do we have that all the K's are directional like the derivatives? e.g. Kxx can have some y meaning - lastly, this can easily be tested and showed in an experiment (i.e. show performance with G and then learned K)
- Is the physical curvature/scale actually estimated? can you compare it to the one from the GT?
- Is the curvature/scale consistent across the cascade? If not, is it getting better/worse compared to GT?
- Are the CDSConv scale features (C1, ... Ck) constrained to be scaled versions of each other? otherwise it's not really a scale estimation, no?
- The use of 2D CNN in the last layer seems to be for performance reasons (not clear from the text), but a comparison to another stage of 3D CNN should provide better results - why not show the impact here?


To summarize, most of my concerns can be addressed with either ad deeper explanation, or preferably an ablation (of using visibility-aggregation  K vs G, removing the last 2D CNN, whether to add the epipolar constraint)


smaller notes:
- "handcrafting" --> "handcrafted"
- Ixx, Iyy, Ixy in eq. 1 are used before definition
- the G<<1 assumption should be better justified
- regarding eq2 - first, why is it intractable? second, there are only 4 convolutions, not 5 (Ixy==Iyx)
- Is the visibility-based aggregation in CDS-MVSNet novel or was it done in prior works? not clear from the text
- are weights shared between cascade scales? (are stages 0,1,2 of Fig 4 identical weights)


**Summary Of The Paper:**

This paper adds upon the recent line of MVSnet works, (specifically CasMVSNet) by explicitly treating the scale needed to reconstruct each part of the scene. This is done by estimating the local curvature of the surface prior to the depth, which is then used to select (in an attention sense) a kernel with an appropriate scale from a set of scaled features in the CDSConv. The curvature estimations is further constrained
to the epipolar line of the relevant source image. The image is processed in a coarse to fine manner similar to CasMVSNet, except the finest resolution which is solved in a (naive?) 2D manner.

**Summary Of The Review:**

The paper is novel, well written, and provides nice SOTA results.
More justification of the design choices can turn it from weak to strong accept.

---

> ### Author Response · Authors · 2021-11-21
> **Detailed explanation for the difference between the proposed learnable curvature and original curvature and providing ablation studies (1/2)**
>
> **[Comment 1]** Replacing Gxx with learned Kxx seems arbitrary: first, using G is quite tractable (only 3 derivatives). second, what promise do we have that all the K's are directional like the derivatives? e.g. Kxx can have some y meaning - lastly, this can easily be tested and showed in an experiment (i.e. show performance with G and then learned K)
>
> **[Answer 1]** We appreciate your comment and agree with the point you made. We want to replace the word “intractable” in our paper by “infeasible,” i.e., using Eq. 2 with the Gaussian filter G is infeasible. We agree that Eq. 2 can still be applied to curvature computation even if the input is a high-dimensional feature map. In this case, we have to compute a curvature map for each feature channel using Eq. 2 and then output the final curvature map by averaging over the channels. However, this solution may produce a noisy estimation because it treats each feature channel equally. In particular, the two neighboring feature vectors are visually similar, but their distance can be large. Therefore, Eq. 2 will produce a high curvature estimation even though the ground truth is low in this case.
>
> For the second question, the kernels K’s are used to handle the high dimensional feature input. Therefore, K’s are high dimensional and learned implicitly from data. As discussed above, G’s average all feature channels; it treats each channel equally. Using K’s instead of G’s can be interpreted as a weighted sum over all channels, which allows extracting information from a specific set of meaningful channels. This approach is an attention mechanism that is a power of deep learning. In summary, to answer the question, K’s might have a directional meaning at some specific channels.
>
> We experimented by using G to justify our theories above, as shown in Table 4 of Appendix A.4. Table 4 shows that using K’s significantly improves the reconstruction quality and computational efficiency performance compared to using G’s.
>
> **[Comment 2]** Is the physical curvature/scale actually estimated? can you compare it to the one from the GT?
>
> **[Answer 2]** Our proposal only approximates the physical curvature by learning it implicitly from data. The main goal is to preserve a property of normal curvature. In particular, the estimated curvature for a pixel X is low when X belongs to an untextured region. In contrast, the curvature is high when it belongs to a rich texture or near-edge region. Our learnable curvature can satisfy this property. However, due to the effect of attention in deep learning, each CDSConv layer tends to focus on scale estimation at some specific regions. To justify these statements, we add Fig. 8 in Appendix A.4 to show the curvature maps extracted from our method and the original method using G’s (Xu et al., 2020).
>
> **[Comment 3]** Is the curvature/scale consistent across the cascade? If not, is it getting better/worse compared to GT?
>
> **[Answer 3]** The curvature is not consistent over cascade stages. Also, it is not consistent in each CDSConv layer. The reason is that the feature vector at each pixel extracted by the current CDSConv layer encodes a different patch scale compared to the feature at the same pixel in the previous CDSConv layer. Although the estimated curvature in each CDSConv layer is different, it still preserves the property of curvature that is low in untextured regions and high in rich texture or near-edge regions. Therefore, it still helps choose a proper scale effectively in our CDSConv module. We have added this information to the manuscript.
>
> (Page 16, Line 14-15): “Fig. 8 also implies that the estimated curvature tends to be higher at the deeper layers because it encodes a larger patch scale.”
>
> **[Comment 4]** Are the CDSConv scale features (C1, ... Ck) constrained to be scaled versions of each other? Otherwise, it's not really a scale estimation, no?
>
> **[Answer 4]** We are sorry for the unclear descriptions. The candidate kernels ${C_1,C_2,…,C_K}$ have different scale/size. Therefore, choosing a kernel among these candidates to extract features is truly a scale estimation. We proposed to choose that kernel by using normal curvature information. To make it clear, we modified the manuscript as follows.
>
> (Page 4, Section 3.2, Line 2): “Given a set of Kconvolutional kernels with different size ${C_1,C_2,…,C_K}$ corresponding to K candidate scales ${\sigma_1, \sigma_2,…,\sigma_K}$, CDSConv aims to select a proper scale for each pixel X.”

---

> ### Author Response · Authors · 2021-11-21
> **Detailed explanation for the difference between the proposed learnable curvature and original curvature and providing ablation studies (2/2)**
>
> **[Comment 5]** The use of 2D CNN in the last layer seems to be for performance reasons (not clear from the text), but a comparison to another stage of 3D CNN should provide better results - why not show the impact here?
>
> **[Answer 5]** We estimated the depth maps for the first three stages $l \in {0,1,2}$ using three depth estimation steps: feature extraction, cost formulation, and cost regularization. For the last stage $l = 3$, we upsample the estimated depth in the previous stage and then perform a refinement with 2D CNN. This 2D CNN did not affect much to the depth accuracy. The reason for using it is to estimate high-resolution depth within a low cost of computation. To analyze the impact of each cascade stage, we also provide an additional experiment (Table 6 in Appendix A.4) to explain our choice for this design.
>
> **[Comment 6]** The G<<1 assumption should be better justified.
>
> **[Answer 6]** We provided a justification for this assumption.
>
> (Page 5, Paragraph 2): “To address the heavy computation issue, we notice that the curvature $curv_{\sigma}$ and derivatives $I_{x^iy^j}$ are proportional to Gaussian kernel $G$, following Eq. 2 and Eq. 3. Moreover, we use a classification network to select the patch scale automatically from the curvature inputs. Therefore, we can perform normalization for the curvatures by rescaling the kernel $G$. We restrict the gaussian kernel in a small range, i.e., $G(X, \sigma)  \ll 1$.
>
> **[Comment 7]** Regarding eq2 - first, why is it intractable? second, there are only 4 convolutions, not 5 (Ixy==Iyx)
>
> **[Answer 7]** As mentioned above, we have changed “intractable” to “infeasible” for more accurate delivery. We pointed out two significant drawbacks of Eq. 2; they are heavy computation and cannot handle the high dimensional feature inputs. We also performed an experiment to validate this, which is shown in Table 4, Appendix A.4. For the second note, there are five convolutions including $I_x,I_y,I_{xx},I_{xy}, I_{yy}$. To make it clear, we revised the manuscript as follows.
>
> (Page 5, Line 2-3): “the computation is heavy because of five convolution operations for computing the derivatives $I_x$,$I_y$,$I_{xx}$,$I_{xy}$, and $I_{yy}$.”
>
> **[Comment 8]** Is the visibility-based aggregation in CDS-MVSNet novel or was it done in prior works? not clear from the text
>
> **[Answer 8]** The visibility-based aggregation is originally proposed by Zhang et al. (2020). However, they only used two-view matching cost volume information and applied a 3D CNN to predict the visibility information. In our visibility prediction, we additionally utilized the curvature prior estimated from the feature extraction network CDSFNet. We then used a 2D CNN for the visibility prediction, which is much more efficient than 3D CNN. We provide a detail explanation about the visibility-based aggregation in Appendix A.4 (Curvature-guided visibility aggregation).
>
> **[Comment 9]** Are weights shared between cascade scales? (are stages 0,1,2 of Fig 4 identical weights)
>
> **[Answer 9]** We applied individual weights for each cascade stage. To make it clear, we revised the manuscript as follows.
>
> (Page 13, Appendix A.2, Line 5): “CDS-MVSNet also uses an independent 3D CNN for each stage to regularize 3D cost volume”
>
> **[Comment 10]** To summarize, most of my concerns can be addressed with either a deeper explanation or preferably an ablation (of using visibility-aggregation K vs G, removing the last 2D CNN, whether to add the epipolar constraint)
>
> **[Answer 10]** In summary, we appreciate the comments of the reviewer so that we can improve our paper. We additionally provide the ablation studies in Appendix A.4 to answer your primary concerns, including the effectiveness of our learnable curvature, the curvature-guided visibility prediction, the role of each cascade stage. Moreover, we analyze the impact of the number of candidate scales ${C_1,C_2,…,C_K}$ in contribution to the overall performance.

---

> ### Comment · Reviewer_6P6T · 2021-11-26
> **Opinion unchanged**
>
> I thanks the authors for the detailed response below. most of my concerns were addressed (I might have been misunderstood for some).
>
> **Small note** - I have tested the authors implementation on some aerial data and it perform on par with CasMVS (not better nor worse), but memory footprint was smaller.
>
> My opinion was positive before the comments and it remains unchanged.

---

> > ### Author Response · Authors · 2021-11-26
> > **Question about the testing**
> >
> > Thank you for your comment.
> > About the testing, did you use our pre-trained model directly or re-training with Aerial dataset? There are some provided pre-trained models which only be trained on a subset of DTU training set for ablation studies in Appendix A.4. Also, we think our proposed model may need more neighboring views for prediction because it can leverage the epipolar geometry.

---

> > ### Author Response · Authors · 2021-11-30
> > **small question**
> >
> > We believe that we already addressed all concerns of the reviewer.
> >
> > However, we find that some evaluated scores such as Correctness, Technical Novelty somehow are unchanged. This may affect the final decision.
> > If the reviewer has any other concerns, please let us know so that we can provide the answer.
> >
> > Thanks for your time,
> >
> > Authors.

---

### Official Review · Reviewer_68iw · 2021-11-02

**Correctness:** 4
**Technical Novelty And Significance:** 3
**Empirical Novelty And Significance:** 2
**Recommendation:** 8
**Confidence:** 5

**Main Review:**

*** Strengths of the paper:

(+) Conceptual novelty.

While the proposed contributions do partially exist in the literature, their combination to address the problem of MVS per se is novel and interesting. This in turn leads to several interesting experimental discussions that are teased early-on in the paper (but lack in their execution, cf. weaknesses below for the missing expected pieces in the evaluation section in particular).

(+) Reproducibility.

The amount of disclosed information regarding the method is fair, and makes uses of the supplementary material by providing comprehensive details regarding the curvature-aware formulation and its differentiable approximation, as well as the code of the proposed method that has also been submitted.

(+) Related Work.

The section (2) is dense, well populated and structured and discusses the main key prior work in the field. While there could have been a broader discussion about the literature overlaps w.r.t aspects of the proposed contributions in difference applicative scopes (eg, the use of 2D curvatures in U-Net based networks), the section is already fairly long (1.2 page long) and serves its purpose relatively well.

(+) Structure, Balance of the Contents.

Overall, the paper is relatively well structured, each section is of pertinent relative size and the text real-estate is well utilized and efficient. To some extent, more qualitative results could make their way to the main paper, instead of predominantly relying on the supplementary material in this regard.


*** Weaknesses of the paper.

- (1) The fairness of the experimental setup is in question.

(a) On the DTU dataset (page 8), as explicitly stated by the authors the proposed method is trained on the union of DTU and Blended-MVS datasets (training splits):

Appendix, A3 page 15:
"Note that we trained the single model using both DTU and Blended datasets
instead of training individually on each dataset as in previous methods (Zhang et al., 2020; Luo
et al., 2020; Sormann et al., 2020). This procedure prevented the model from biasing to each dataset
in the evaluation step; it guaranteed the generalized property of model."

This is a major issue because the reported performance figures of the competition in Table 1 are being trained on the conventional train split of DTU, without additions.

(b) Appendix, A3 page 15:  It is factually not true that (Zhang et al., 2020; Luoet al., 2020; Sormann et al., 2020) perform training on DTU+Blended-MVS to test on DTU:

(i) Zhang et al., 2020:
The experiments on DTU and Blended-MVS are performed by training on the respective conventional training splits, NOT on both training sets simultaneously. This is explicitly stated by the authors Page 7, sec 4.1 of their main paper.

(ii) Luo et al., 2020:
Their model is trained and tested solely on DTU, using the conventional splits (sec 5.2 in their main paper).

(iii) Sormann et al., 2020:
Experiments on DTU: the training is performed solely on the training split of the dataset.
For Tanks and Temples and ETH3D (which is not considered in the proposed paper), the authors additional fine-tune on Blended-MVS on top of the conventional DTU training. This can be found in their main paper, sec 5.1.2 and sec 4 at length.

(c) Regarding the provided experiments on Tanks and Temples (ToG 2017), Table 3:

It is a common practice in the field to train on either DTU  with or without fine-tuning on Blended-MVS (eg, resp. BP-MVS, AttMVS), not by training on the union of both dataset as it is done for the proposed method (Appendix A3).

In particular, given that AttMVS is the closest-performing competition w.r.t the proposed method, why not provide experiments with the same training setup / data?

This was an easy way to directly address the relative positioning against the most competitive player in the field. Chosing a different setup can be interpreted as a means to hide potential red flags.

- (2) Certain claims and justifications are either ambiguous, inaccurate or even misleading.

This is the case in particular regarding the evaluation part above, and the pieces of justification that are specified in the appendix. This agin, is a relatively major concern as is.

- (3) No ablation studies are provided.

Given the sequential pipeline nature of the proposed contributions, ie, feature extraction, approximate curvature priors, 3D cost volume aggregation and regularization, it is highly expected to provide a comprehensive ablation study (ie, activating/removing pieces of the pipeline) to help understand and assess the relative contribution of each individual piece / ingredient, in particular given:

(a) It is a common practice in the field and within the realm for this particular literature (eg, BP-MVS, AttMVS, or even at BMVC which is a venue which much less text real-estate: Zhang et al. 2020).

(b) This is all-the-more problematic given (1) above, that puts an even greater weight on the ablation analysis and the lack thereof.

(c) Additionally, certain design choices and hyperparameters are not experimentally challenged to support their respective values are (at least near-) optimal, eg, \lambda_1 and \lambda_2 in sec 4.2, page 13: the number of different scales / number of stages that are typically considered.

- (4) Missing prior work in the comparative evaluation.

This is relatively important given the proximity, performance-wise, w.r.t the proposed method, eg, (Sormann et al. 2020) BP-MVS (overall perf: 0.327 vs. 0.315).

- (5) Relative positioning, performance-wise, is unclear.

This is a direct implication stemming from (1)+(3) above. The reported performance figures are hence uninformative and do not allow to properly assess the relative positioning of the proposed approach w.r.t existing work, and hence, the validate its usefulness.

- (6) Readability.

To a lesser extent, the text as it currently stands, is still in a relatively rough, tough to digest state. There are numerous grammar issues and typos (a few example below). However and overall, the main ideas and key claims do not suffer substantially from the language-related issues and the articulations of the main ideas is often preserved.

A few examples with suggestions for improvements:
- abstract l3: by introducing a skilled design to cost formulation -> by designing aggregated 3D cost volumes and their regularization.
- abstract: our final MVS architecture -> our resulting MVS architecture.
- abstract: Moreover -> As a result,
- abstract: can process the high resolution with faster run-time -> can process higher resolution inputs within faster runtimes
- introduction, first paragraph: an object's scale varies extremely in images -> in presence of high/significant scale variations of objects in images
- throughout: dynamic scale networks -> scale-adaptive networks

**Summary Of The Paper:**

The paper proposes a novel Deep Neural Net to address the problem of Multi-View Stereo (MVS) by introducing several claimed novel ingredients, ie, (i) CDFSNet: a scale-adaptive feature extraction network that takes into account (ii) an approximation of image-based 2D curvatures in a U-Net based network;
(iii) CDS-MVSNet: is the actual reconstruction network that consider multiple resolutions and input feature scales into an aggregated, regularized multi-view cost-volume.

The end-to-end sequential pipeline forming the proposed contribution is evaluated on public, popular standard benchmarks in the field, showing promising results, despite the fairness of the experimental setup being in question (as detailed below in the review).

**Summary Of The Review:**

*** Summary and justification of the initial rating.

Overall, the paper presents very interesting and relatively new ingredients to address the problem of Multi-View Stereo using supervized learning. The presentation of the contents, despite a few substantial issues, also lies in the list of positives.

However, as it currently stands, the proposed package suffers from major drawbacks in the evaluation department, as detailed above in the main review. In particular:

(i) The proposed experimental comparisons are performed in a fair setup, for the most part and regarding the most important dataset / set of experiments.

(ii) As an extension to (i), there are key claims, references and justifications that are inaccurate to say the least, that could easily mislead the reader.

(iii) Given the nature of the proposed contributions forming a sequential pipeline, (ie: feature extraction, approximate curvature priors, 3D cost volume aggregation and regularization) it is highly expected to provide a comprehensive ablative study to analyze, motivate and qualtify the need of each individual ingredient. Such experiments have not been proposed.

As a result, the proposed package is very interesting and the results promising, but the execution of the evaluation part do not allow to validate the main claims and novelties so far.

---

> ### Author Response · Authors · 2021-11-21
> **Explanation for the original design of experiments, designing more experiments for the fairness, providing ablation studies, and revising the ambiguous sentences in the paper (1/3)**
>
> **[Comment 1]** ***(1)*** The fairness of the experimental setup is in question. ***(a)*** On the DTU dataset (page 8), as explicitly stated by the authors the proposed method is trained on the union of DTU and Blended-MVS datasets (training splits). This is a major issue because the reported performance figures of the competition in Table 1 are being trained on the conventional train split of DTU, without additions.
>
> **[Answer 1]** We appreciate your comment and agree with the point you made. The feature extraction of CDSFNet heavily depends on the epopilar geometry; therefore, we require a training dataset with diverse camera trajectories. However, the DTU dataset is obtained from the same camera trajectory for all 118 scenes. The DTU dataset alone is not sufficient to effectively train our model. Therefore, we additionally used the BlendedMVS dataset to get the images captured from diverse camera trajectories.
>
> To provide fair experimental results, we additionally evaluated our model trained only on the DTU training dataset. We applied the same setups and hyperparameters for the evaluation as before. The accuracy, completeness, and overall performance are appended to Table 1. Our method still outperforms the state-of-the-art baselines and approximates the results trained on both DTU and BlendedMVS datasets.
>
> **[Comment 2]** ***(1)*** ***(b)*** Appendix, A3 page 15: It is factually not true that (Zhang et al., 2020; Luoet al., 2020; Sormann et al., 2020) perform training on DTU+Blended-MVS to test on DTU
>
> **[Answer 2]** We are sorry for the unclear description. We agree that the written sentence "Note that we trained the single model using both DTU and Blended datasets instead of training individually on each dataset as in previous methods (Zhang et al., 2020; Luo et al., 2020; Sormann et al., 2020)" is ambiguous. In this sentence, we mean that the previous methods (Zhang et al., 2020; Luo et al., 2020; Sormann et al., 2020) trained individually on each dataset, whereas we trained a single model using both DTU and Blended datasets. We revised this sentence as "We provided two experimental scenarios.  First, we train the model on the DTU training set and evaluate the DTU test set. This pre-trained model is then fine-tuned on BlendedMVS dataset for 10 epochs with a learning rate of $10^{-4}$ to evaluate on Tanks & Temples. Second, we train a single model on both DTU and BlendedMVS datasets and use this model for evaluation on both DTU and Tanks & Temples."
>
> **[Comment 3]** ***(1)*** ***(c)*** Regarding the provided experiments on Tanks and Temples (ToG 2017), Table 3: It is a common practice in the field to train on either DTU with or without fine-tuning on Blended-MVS (eg, resp. BP-MVS, AttMVS), not by training on the union of both dataset as it is done for the proposed method (Appendix A3).
>
> **[Answer 3]** We understand your concern. However, in our case, we trained on both datasets as follows
>
>     for epoch in 1…num_epochs:
>         for batch in DTU:
>             update weights
>         for batch in Blended:
>             update weights
>         decrease learning_rate
>
> The method that trains on DTU and then fine-tunes on BlendedMVS such as BP-MVS can be presented as follows
>
>     for epoch in 1…num_epochs:
>         for batch in DTU:
>             update weights
>         decrease learning_rate
>     set learning_rate a small value
>     for epoch in 1…num_finetuning_epochs:
>         for batch in BlendedMVS:
>             update weights
>         decrease learning_rate
>
> Compared to the fine-tuning approach above, there will be no considerable difference in performance in our approach. We did not shuffle DTU and BlendedMVS and then sampled the batch for updating the weights. We updated the network weights on DTU and BlendedMVS sequentially.
>
> To verify this, we performed the fine-tuning and then submitted the reconstructed models to the Tanks & Temples benchmark. The performance was not too different in comparison to the performance by training on DTU and Blended simultaneously. However, the performance was degraded because we only fine-tuned with a small number of epochs. We found that the loss can still be optimized when increasing the number of epochs. We added the results to Table 3 for reference.

---

> ### Author Response · Authors · 2021-11-21
> **Explanation for the original design of experiments, designing more experiments for the fairness, providing ablation studies, and revising the ambiguous sentences in the paper (2/3)**
>
> **[Comment 4]** In particular, given that AttMVS is the closest-performing competition w.r.t the proposed method, why not provide experiments with the same training setup and data?
>
> **[Answer 4]** Compared to the other state-of-the-art methods, AttMVS applied different training and evaluation setups, as follows:
> - First, AttMVS used the refined DTU dataset by generating more accurate ground-truth depths for training while most of the other methods, including ours, used the DTU dataset preprocessed by Yao et al. (2018) for MVSNet.
>
> - Second, AttMVS applied a higher number of depth hypotheses for training on DTU, 256. The other methods used less than that; for example, CasMVSNet (Gu et al., 2020) and VisMVSNet (Zhang et al., 2020) applied 192 and 128 depth planes, respectively. Our training setup is most similar to CasMVSNet.
>
> - Finally, AttMVS proposed an improved point cloud generation strategy presented in Section 4 of their paper. Most other methods such as MVSNet, CasMVSNet, CVPMVSNet, VisMVSNet, and ours applied the same point cloud generation strategy. On the other hand, AttMVS further performed the point cloud refinement process by optimizing a target function. The refinement remarkably contributes to the quality of reconstructed 3D models.
>
> The source code and pre-processed dataset of AttMVS are not available, but the re-implementation of AttMVS is out of scope in our paper. For these reasons, we did not follow the setups of AttMVS. Excepting training on DTU and Blended simultaneously, we followed the general setups of recent works such as MVSNet, CasMVSNet, CVPMVSNet, VisMVSNet, etc.
>
> In summary, we agree that training on the union of DTU and Blended is a different setup. However, recent studies such as AttMVS (Luo et al., 2020), BP-MVSNet (Sormann et al., 2020), and Vis-MVSNet (Zhang et al., 2020) applied their own setups for training to ensure that their models have the best performance. For example,
>
> - AttMVS designed the setups as mentioned above.
>
> - BP-MVSNet integrated the validation set with 18 scenes to the training set of DTU and then performed model training on the integrated dataset; this can be found in Section 4 in their paper. The other methods, including ours, generally trained on the training set, validated their models on the validation set, and finally reported the evaluation on the test set of DTU. For Tanks & Temples, BP-MVSNet fine-tuned the pretrained model on DTU by re-training the model on the BlendedMVS dataset with a learning rate of $10^{-4}$.
>
> - VisMVSNet applied a noise filtering strategy for point cloud generation by using all probability maps at all stages of coarse-to-fine architecture (Section 4.1 in their paper). On the other hand, previous methods such as CasMVSNet, CVP-MVSNet, UCSNet used the probability map only at the last stage for filtering.
>
> Therefore, we also keep the original results of our model, which are produced by training the models on two datasets simultaneously. We also provide the new results by applying different dataset setups, as follows:
>
> - We trained our model only on the training set of DTU and evaluated its performance on the test set. This result was updated in Table 1.
>
> - We fine-tuned this model pre-trained on DTU using the BlendedMVS dataset with 10 epochs and a learning rate of $10^{-4}$ to evaluate the Tanks & Temples benchmark performance. We added this result to Table 3.
>
> In both cases, our method still achieved good performance in all benchmarks. This demonstrates the effectiveness of our proposed method.
>
> **[Comment 5]** ***(2)*** Certain claims and justifications are either ambiguous, inaccurate or even misleading. This is the case in particular regarding the evaluation part above, and the pieces of justification that are specified in the appendix. This again is a relatively major concern as is.
>
> **[Answer 5]** We fixed the ambiguous sentences and added more evaluations as you suggested. As explained above, training on both datasets can be an option.

---

> ### Author Response · Authors · 2021-11-21
> **Explanation for the original design of experiments, designing more experiments for the fairness, providing ablation studies, and revising the ambiguous sentences in the paper (3/3)**
>
> **[Comment 6]** ***(3)*** No ablation studies are provided. Given the sequential pipeline nature of the proposed contributions, ie, feature extraction, approximate curvature priors, 3D cost volume aggregation and regularization, it is highly expected to provide a comprehensive ablation study (ie, activating/removing pieces of the pipeline) to help understand and assess the relative contribution of each individual piece / ingredient, in particular given: ***(a)*** It is a common practice in the field and within the realm for this particular literature (eg, BP-MVS, AttMVS, or even at BMVC which is a venue which much less text real-estate: Zhang et al. 2020). ***(b)*** This is all-the-more problematic given (1) above, that puts an even greater weight on the ablation analysis and the lack thereof. ***(c)*** Additionally, certain design choices and hyperparameters are not experimentally challenged to support their respective values are (at least near-) optimal, eg, \lambda_1 and \lambda_2 in sec 4.2, page 13: the number of different scales / number of stages that are typically considered.
>
> **[Answer 6]** Thank you for your helpful suggestions. We added the ablation studies in Appendix A.4. In the ablation studies, we addressed the following issues:
>
> - Effectiveness of learnable curvature: We validated our proposed learnable curvature by comparing it with the original version proposed by Xu et al. (2020). We replaced our curvature estimation in Eq. 5 with the original in Eq. 1 and then performed the same training and evaluation setups. As shown in Table 4, the proposed learnable curvature boosts the performance significantly compared to the original curvature. It also improves efficiency in terms of runtime and memory consumption.
>
> - Curvature-guided visibility aggregation: We analyzed the effectiveness of curvature-guide visibility prediction on the DTU dataset. This visibility prediction improves the depth estimation accuracies.
>
> - Analysis of CDSFNet: To deeply understand our proposed feature network CDSFNet, we removed the cost regularization step in our CDS-MVSNet pipeline and then changed the number of candidate scales in each CDSConv layer. By designing the experiment like this, we can find out how CDSFNet affects feature matching and improves performance when increasing the network complexity. Table 5 shows the accuracy of estimated depth when increasing the number of candidate scales from 1 to 4. We observed that CDSFNet achieved better performance when using more candidate scales.  However, it suffered from heavy computation. Therefore, we chose the design with two or three scales for each CDSConv, shown in CDSFNet architecture (Fig. 5), to guarantee depth accuracy and computational efficiency.
>
> - Analysis of the cascade structure: Finally, we investigated how much the estimated depth is improved after each cascade stage. Combing with the evaluations in Table 1, 3 and Fig. 3, we demonstrated that our design could guarantee both reconstruction quality and computational efficiency.
>
> **[Comment 7]** ***(4)*** Missing prior work in the comparative evaluation. This is relatively important given the proximity, performance-wise, w.r.t the proposed method, eg, (Sormann et al. 2020) BP-MVS (overall perf: 0.327 vs. 0.315).
>
> **[Answer 7]** The results of BP-MVSNet are updated in Table 1 and Table 3 for DTU and Tanks & Temples, respectively. However, as mentioned in **Answer 4**, the BP-MVSNet’s model is trained on both the training and validation sets of DTU. The model is then fine-tuned on the BlendedMVS dataset for the Tanks & Temples benchmark. Although BP-MVSNet achieves an approximated performance compared to our method on the DTU dataset, its performance on Tanks & Temples is lower than ours with a large margin.
>
> **[Comment 8]** ***(5)*** Relative positioning, performance-wise, is unclear. This is a direct implication stemming from (1)+(3) above. The reported performance figures are hence uninformative and do not allow to properly assess the relative positioning of the proposed approach w.r.t existing work, and hence, the validate its usefulness.
>
> **[Answer 8]** We appreciate your comment and agree with the point you made. We added more evaluations in Tables 1 and 3 and updated the qualitative comparisons in Figs. 9 and 10.
>
> **[Comment 9]** ***(6)*** Readability. To a lesser extent, the text as it currently stands, is still in a relatively rough, tough to digest state. There are numerous grammar issues and typos (a few example below). However and overall, the main ideas and key claims do not suffer substantially from the language-related issues and the articulations of the main ideas is often preserved.
>
> **[Answer 9]** We appreciate your comment. We revised the ambiguous sentences and fixed them.

---

> ### Comment · Reviewer_68iw · 2021-11-29
> **Improved rating from [Reject (3)] to [Accept (8)]**
>
> Dear All,
>
> *** TL;DR:
> [Reject (3) -> Accept (8)]
>
> *** Detailed feedback:
> Thanks to the authors' substantial efforts to improve the submitted contents, I do bump my initial rating to a plain accept.
>
> In particular, the initial submission package suffered from three root-causes for concern, ie:
>
> - (1) The comparative evaluation not being carried in a fair, transparent setup in comparison to prior art.
>
> - (2) Major issues also related to the ablation study, in particular, the lack thereof, to illustrate the relative performance by enabling / desabling each individual contributed component to the overall proposed method.
>
> - (3) And finally to a lesser extent, perceivable readability issues.
>
> All of the remaining issues in my initial review stemmed from these three main root-causes.
>
> *** As a result of the updated contents:
>
> - (1) has been convincingly addressed notably by adding experiments to Table 1 and Table 3 in the main paper which are pivotal additions, that now support the crucial claim the paper makes regarding the relative positioning of the proposed system to prior work, performance-wise.
>
> - (2) The addition of an explicit and transparent ablation study in the appendix section A.4, this issue is also now convincingly addressed and validates the relative usefullness of the main modular ingredient in the proposed contribution: the integration of the view-depdendant surface curvature in the proposed neural net.
>
> - (3) Readability-wise, the overall packaged is substantially more readable and very reasonably within cam-ready reach. The main key ambiguous statements have been addressed and the suggestions of several reviewers with this regard have been taken into account.
>
> - (3) *Yet*, I strongly invite the authors to do one modification to Tables 1--3 in the cam-ready by providing the information (per method) regarding the individual Train/Validation/Test splits for each individual experiment to help the reader making sense of each of the reported performance vs. experimental setup.
>
> - Other sizeable improvements have also been made such as the valuable additions of more qualitative results in the appendix.

---

> > ### Author Response · Authors · 2021-11-30
> > **About updating the rating in your official review**
> >
> > Thanks for your feedback.
> > Can you please update the final rating in the official review? We find that the rating is still 3 in your official post.

---

### Official Review · Reviewer_Qeua · 2021-11-02

**Correctness:** 3
**Technical Novelty And Significance:** 3
**Empirical Novelty And Significance:** Not applicable
**Recommendation:** 8
**Confidence:** 4

**Main Review:**

Strength:
- The rationale behind finding the right scale to disambiguate matching is well written and quite reasonable. It would be good to pinpoint though, the exact example that suffers from the matching ambiguity issue, i.e. highlight some pixels in the image pairs about the wrong feature association.
- The quantitative results show the method is competitive.
- Run time is also lower when image resolution increases, compared with CasMVSNet and others.

Weakness:
- The writing can be improved. For example, the learnable normal curvature part is a bit hard to read.
- It’d be good to add some ablation studies to show the delta brought by CDSConv and the MVS architectural changes compared with the baseline network, i.e. CasMVSNet.


Feedback:
- For the Tanks&Temple dataset, did you also evaluate the method using high-res images on the advanced cases? I found that the method ranks high on the intermediate cases but not on the advanced cases.
- Is the major deduction of running time a consequence of only computing depths of half-resolution images and then upsample to the original resolution?
- Add legend to the left image of Figure 3. Why is the GPU memory significantly lower when resolution is increased? It’s even lower compared with lower-res images, for ‘ours’.




**Summary Of The Paper:**

This paper proposes a method named CDSFNet for the multi-view stereo problem. The network is composed of curvature-guided dynamic scale convolution (CDSConv) layers to first estimate approximate normal curvature at different candidate scales, and then do a classification to find the optimal scale. The module is then embedded with cascade structure of CasMVSNet, to reduce the matching ambiguity by considering the proper pixel scales. The method is evaluated on DTU, BlendedMVS and Tanks & Temples datasets, which archives state-of-the-art performance compared with other related methods.


**Summary Of The Review:**

I recommend Accept for this paper since the strengths outweigh the weaknesses. The paper is generally written well with good performance. One thing that can be improved as I mentioned above is more ablation experiments.

---

> ### Author Response · Authors · 2021-11-21
> **Explanation for the evaluation of Tanks&Temples and the evaluation of computational efficiency. An ablation study is provided to justify the effectiveness of proposed methods.**
>
> **[Comment 1]** It’d be good to add some ablation studies to show the delta brought by CDSConv and the MVS architectural changes compared with the baseline network, i.e. CasMVSNet.
>
> **[Answer 1]** We appreciate your comment and agree with the point you made. We additionally provide an ablation study in Appendix A.4 (analysis of CDSFNet and the cascade structure). This section analyzes the effectiveness of several proposed modules, including learnable normal curvature, curvature-guided for visibility prediction, number of candidate scales in CDSFNet, and the number of stages in cascade MVS structure.
>
> **[Comment 2]** For the Tanks&Temple dataset, did you also evaluate the method using high-res images on the advanced cases? I found that the method ranks high on the intermediate cases but not on the advanced cases.
>
> **[Answer 2]** We evaluated the advanced set of Tanks & Temples. However, similar to most learning-based methods trained on DTU and BlendedMVS datasets, our method achieved only an average ranking in this benchmark. The main reason is due to the complex indoor scenes in the advanced dataset. Moreover, our paper is already long enough with dense experiments and analysis. Therefore, we did not include this evaluation in our paper. The improvement for the advanced dataset can be addressed by re-training our model by using an additional indoor dataset.
>
> **[Comment 3]** Is the major deduction of running time a consequence of only computing depths of half-resolution images and then upsample to the original resolution?
>
> **[Answer 3]** Yes, our method can predict highly accurate depth maps even at half-resolution because of the robustness of our feature extraction CDSFNet. Therefore, we only need to maintain that accuracy at the full resolution by upsampling and then performing a depth refinement step. This approach can guarantee both reconstruction quality and computational efficiency for high-resolution MVS. To explain this, we have revised the manuscript as follows.
>
>  (Page 17, Subsection “Analysis of CDSFNet and the cascade structure”, Paragraph 2, Line 3): “The last stage l = 3 is used for depth upsampling; it outputs the full-resolution depth while maintaining the same depth accuracy compared to the previous stage. Suppose this stage is implemented by the three-step MVS depth estimation pipeline mentioned in Section 4.1, the entire MVS network suffers from a considerable computation and cannot fit into a limited GPU memory. Therefore, we only apply the depth refinement with a 2D CNN to the last stage for efficiency in time and memory.”
>
> **[Comment 4]** Add legend to the left image of Figure 3. Why is the GPU memory significantly lower when resolution is increased? It’s even lower compared with lower-res images, for ‘ours’.
>
> **[Answer 4]** We have updated Fig. 3 with a legend. About the reduction of GPU memory, based on the observations from Fig. 3, we notice that memory consumption starts to drop when the runtime increases non-linearly. This means that the executing program scarifies a small amount of time to finish some parallel computations and then release memory. The reason is due to a gap of computation between the last stage and previous stage when the image is at a high resolution. In particular, the last stage used a 2D CNN while the previous stage used a 3D CNN. At the low resolution, the difference of computation between these stages is small. However, at the high resolution, the computation in previous stage is heavier than the last stage. There will be a delay in time to pass the depth estimated from the previous stage to the last stage. In the delay time, the memory might be released significantly. Therefore, the memory is dropped at high resolution in our method. We have revised the manuscript for this explanation as follows.
>
> (Page 9, Subsection “Run-time and memory”, Line 6): “However, we noticed that the memory consumption of our method was dropped from the resolution of 80% while our run-time was slightly increased. This indicates that our method can release a significant amount of GPU memory by only sacrificing a small amount of run-time.”.

---

> > ### Comment · Reviewer_Qeua · 2021-11-26
> > **My final rating is accept (Unchanged)**
> >
> > Thanks authors for the detailed response. I read the comments and found it convincing enough at least for my concerns. Also thanks for the follow-up experiments and revisions to the text. I think they are great additions to be added in the appendix. I keep my initial rating and think this is a good paper to appear in ICLR.

---

### Official Review · Reviewer_aAob · 2021-11-03

**Correctness:** 4
**Technical Novelty And Significance:** 3
**Empirical Novelty And Significance:** 2
**Recommendation:** 6
**Confidence:** 3

**Main Review:**

This paper is basically well written and has several strengths.
+ Using dynamic scale convolution for learning-based multi-view stereo is new.
+ Using the curvature of the target scene for determining kernel scale would be adequate.
+ Well experimented (but only with learning-based methods)


My main concern for this paper is about the main contribution.
Using curvature information to select the patch scale is used in an MVS method (Xu et al. (2020)).
Although the method does not use deep learning, it limits the technical contribution of this paper.

In the related work section, this paper claims that the proposed method should have an advantage of performance compared with Xu et al. (2020) because they do not rely on the handcrafting features.
However, the claim was not confirmed through the experiments.

*** After discussion phase ***
During the discussion phase, the authors added a comparison of the proposed method with Xu et al. (2020).
From the comparison, indeed the proposed approach shows advantages for both accuracy and computation time.

Therefore, compared with the initial rating, I changed my rating to a positive one.
A dynamic-scale method using curvature has been proposed so far, somehow limiting the novelty of this work.
Meanwhile, developing a learning-based approach would be beneficial.

**Summary Of The Paper:**

This paper proposes a method for deep-learning-based multi-view stereo. The contribution of the paper would be to adjust the scale of convolutional kernels according to the curvature of the target scene. Experiments show the proposed method (slightly) outperforms the existing deep-learning-based multi-view stereo methods.

**Summary Of The Review:**

*** For the initial review***
I am concerned about the effectiveness of the main claim of this paper because it is not confirmed through the experiment, i.e., it is not compared with the most important existing method.

***As for the final rating,***
I acknowledged the comparison with the most important method and confirmed the advantage of the proposed method.

---

> ### Author Response · Authors · 2021-11-21
> **Explanation of technical contributions and designing more experiments for justification**
>
> **[Comment 1]** My main concern for this paper is about the main contribution. Using curvature information to select the patch scale is used in an MVS method (Xu et al., 2020). Although the method does not use deep learning, it limits the technical contribution of this paper.
>
> **[Answer 1]** Thank you for your review. We agree that the existing work already utilized the curvature information to select the pixel-level scale (Xu et al., 2020). However, compared to this work, our proposed method has several main contributions as follows
> -	We proposed an improved version for curvature computation, namely learnable curvature, which is fast and applicable to a deep neural network.
> -	Our method automatically selected a proper scale among the candidates by using a classification network. Meanwhile, the previous work used a threshold of 0.01 for curvature to select a scale, which degrades the performance. Our scale selection can adapt to epipolar geometry, which is not analyzed in previous work. This property is already shown in Figures 2 and 6 in our paper.
> -	We designed a robust feature extraction network for multi-view stereo (MVS). We also applied the estimated curvature to predict visibility information for cost aggregation, this significantly enhances the depth estimation accuracies.
>
> We detail the contributions of our work compared to Xu et al. (2020) in Appendix A.4 (Effectiveness of learnable curvature).
>
> **[Comment 2]** In the related work section, this paper claims that the proposed method should have an advantage of performance compared with Xu et al. (2020) because they do not rely on the handcrafting features. However, the claim was not confirmed through the experiments.
>
> **[Answer 2]** To verify the statements above, we additionally perform the ablation studies in Appendix A.4 (Effectiveness of learnable curvature). Table 4 in Appendix A.4 presents the results of our proposed learnable curvature compared to the original curvature by Xu et al. (2020).  We can summarize the results in Table 4 as follows
>
> | Methods             | Prec. 2mm | Prec. 4mm | MAE (mm)| Accuracy (mm) | Completeness (mm) | Overall (mm) | Time (s) | Memory (MB) |
> |---------------------|-----------|-----------|---------|---------------|-------------------|--------------|----------|-------------|
> | original curvature  | 74.26     | 84.89     | 6.84    | 0.378         | 0.315       | 0.347   | 0.539| 7993   |
> | learnable curvature | **76.23** | **85.92**     | **5.89**    | **0.372**         | **0.305** | **0.339** | **0.421** | **4389** |
>
> Our method significantly improves the performance in terms of depth accuracy, reconstruction quality, runtime, and memory consumption compared to the method with original curvature. We also provide a qualitative comparison in Fig. 8. The original curvature estimation degrades the performance because it cannot handle the high-dimensional features. It treats each feature channel equally when computing the derivatives with Gaussian filters. Therefore, it may produce a high curvature estimation even though the neighboring feature vectors are visually similar. On the other hand, the learnable curvature consistently preserves the properties of normal curvature concerning the untextured and rich-textured regions in every layer. Furthermore, our learning-based estimation takes advantage of the attention mechanism (shown in the red circles in Fig. 8), which improves the completeness of reconstruction.
>
> We also provide the effectiveness of the curvature-guided visibility aggregation. As shown in Table 4, the curvature-guided visibility aggregation boosts the performance without more heavy computation.
>
> In the main experiments, we evaluate the performances on DTU and Tanks & Temples datasets, the most commonly used for MVS evaluation. The original study of the MARMVS method (Xu et al., 2020) did not provide an evaluation on these datasets. Re-implementing the whole MARMVS pipeline is out of scope in this paper. However, we additionally compare our method to the other traditional MVS methods, such as COLMAP, Gipuma (shown in Table 1), ACMM, and ACMP (updated in Table 3). Our method outperformed the state-of-the-art methods on DTU and achieved a high ranking on the leaderboard of Tanks & Temples.

---

> > ### Comment · Reviewer_aAob · 2021-11-29
> > **Final rating**
> >
> > My main concern at the initial review was the lack of comparison with Xu et al. (2020), which could be the most critical existing work.
> >
> > During the discussion phase, the authors added the comparison.
> > From the comparison, indeed, the proposed approach shows advantages for both accuracy and computation time.
> >
> > Therefore, compared with the initial rating, I changed my rating to a positive one:
> > A dynamic-scale method using curvature has been proposed so far, somehow limiting the novelty of this work.
> > Meanwhile, developing a learning-based approach would be beneficial.

---

### Author Response · Authors · 2021-11-21
**Thanks for the comments and suggestions of the reviewers**

Dear reviewers and editors,

We appreciate your careful reading of our manuscript and insightful comments. We believe that we have addressed all of the reviewers’ concerns and suggestions. Below, please find our responses to all of your comments. Our manuscript has been revised accordingly; these revisions have greatly improved the clarity of our paper. We summarize the key changes as follows

- updated the details of experimental setups in Appendix A.3
- added more experimental results and the baselines in Table 1 and Table 3, updated the qualitative results on DTU and Tanks & Temples in Appendix A.5
- added ablation studies for the detailed explanation of contributions which are shown in Appendix A.4.
- added more explanation in Section 3.2, subsection "learnable normal curvature"
- fixed typos, ambiguous sentences

We also provide source code and all pretrained models in the supplementary materials for reproducibility. We hope our manuscript is now suitable for publication. Thank you very much.

Sincerely Yours,

---

### Decision · Program_Chairs · 2022-01-20

**Decision:**

Accept (Poster)

**Comment:**

All reviewers recommended accept after discussion. I am happy to accept this paper.